# Room temperature exciton–polariton Bose–Einstein condensation in organic single-crystal microribbon cavities

Ji Tang[1,2], Jian Zhang[3], Yuanchao Lv[1], Hong Wang[1,2], Fa Feng Xu [1,2], Chuang Zhang[1,2], Liaoxin Sun[3], Jiannian Yao[1,2] & Yong Sheng Zhao [1,2✉]

Exciton–polariton Bose–Einstein condensation (EP BEC) is of crucial importance for the development of coherent light sources and optical logic elements, as it creates a new state of matter with coherent nature and nonlinear behaviors. The demand for room temperature EP BEC has driven the development of organic polaritons because of the large binding energies of Frenkel excitons in organic materials. However, the reliance on external high-finesse microcavities for organic EP BEC results in poor compactness and integrability of devices, which restricts their practical applications in on-chip integration. Here, we demonstrate room temperature EP BEC in organic single-crystal microribbon natural cavities. The regularly shaped microribbons serve as waveguide Fabry–Pérot microcavities, in which efficient strong coupling between Frenkel excitons and photons leads to the generation of EPs at room temperature. The large exciton–photon coupling strength due to high exciton densities facilitates the achievement of EP BEC. Taking advantages of interactions in EP condensates and dimension confinement effects, we demonstrate the realization of controllable output of coherent light from the microribbons. We hope that the results will provide a useful enlightenment for using organic single crystals to construct miniaturized polaritonic devices.

[1] Key Laboratory of Photochemistry, Institute of Chemistry, Chinese Academy of Sciences, Beijing, China. [2] University of Chinese Academy of Sciences, Beijing, China. [3] State Key Laboratory of Infrared Physics, Shanghai Institute of Technical Physics, Chinese Academy of Sciences, Shanghai, China.
✉email: yszhao@iccas.ac.cn

Exciton–polaritons (EPs) arising from strong coupling between excitons and photons can form a macroscopic condensate via bosonic final-state stimulation into the ground state—a process known as Bose–Einstein condensation (BEC) of EPs[1,2]. The formed EP Bose–Einstein condensates, exhibiting spontaneous coherences and large nonlinearities, attract great attention for the implementation of ultralow-threshold lasers[3–5], high-speed switches[6–8], and all-optical logic gates[9]. Efforts in EP BEC were initially focused on inorganic semiconductor quantum well microcavities, most of which suffer from limitations imposed by cryogenic temperature operation and extreme fabrication conditions[10–12]. Such limitations have motivated the search for new materials with ease of fabrication for room temperature EP BEC[13], and instigated the development of strong coupling in organic materials[14,15]. This is because organic materials possess tightly bounded Frenkel excitons with high binding energy, making them an ideal system for achieving EP BEC at room temperature.

Room temperature exciton–polariton Bose–Einstein condensation (EP BEC) and related phenomena such as polariton lasing[16,17], superfluidity[18], and nonlinear amplification[19] have been demonstrated with a broad range of organic materials including small molecules[20–22], polymers[23], and biomolecules[24]. In these works, the organic materials were sandwiched between two distributed Bragg reflectors (DBRs) to form a high-finesse microcavity for strong exciton–photon coupling. Such planar (two-dimensional) microcavities tend to have large lateral device footprints and complications to guide EP fluids[9,19], which restricts the development of organic polaritonic devices with compactness and integrability. Recent progresses on cavity-free organic polariton systems, such as molecular crystals[25] and topological plexitons[26,27], offer a strategy to address this issue by realizing BEC in structures with diverse geometries. Organic micro/nanostructures can form natural microresonators[28], making them an alternative to achieve strong coupling in low-dimensional geometries without having to rely on external microcavities[29,30]. However, the realization of EP BEC in organic micro/nano-structures without DBRs until now has been prevented by their limited cavity quality factor ($Q$) and the resulted small exciton–photon coupling strength. Increasing the densities of Frenkel excitons can compensate for the decrease in the cavity $Q$[31], which offers a possibility to enhance exciton–photon coupling strength in micro/nano-structures by selecting organic materials with high optical transition probability. Moreover, assembling these organic molecules into single-crystalline micro/nano-structures would further help to realize room-temperature EP BEC, because single crystals exhibit highly packed densities of organic molecules and low non-radiative losses[32], which is favorable for maintaining large numbers of Frenkel excitons.

Here, we demonstrate room-temperature polariton Bose–Einstein condensation in organic single-crystal microribbon cavities, in which the large densities of stable Frenkel excitons attainable from the organic molecules lead to efficient strong exciton–photon coupling. The self-assembled microribbons with regular shape function as waveguide Fabry–Pérot (F–P) microcavities, where the orderly packed organic molecules provide Frenkel excitons with large transition dipole moments, which can strongly couple with microcavity photons at room temperature. The efficient strong exciton–photon coupling forms high-density polaritons, which undergo Bose–Einstein condensation by stimulated scattering into the polariton ground state. Thanks to the manipulability of polariton condensates, we further demonstrated control over the output of coherent light based on the repulsive interaction between polaritons and the reservoir of uncondensed excitons. The results not only reveal the important role of molecular packing for polariton condensation in organic single crystals but also suggest great superiority of organic single-crystal micro/nano-structures towards polariton Bose–Einstein condensation for on-chip photonic circuit applications.

## Results and discussion

**Strong coupling in the organic single-crystal microribbons.** N, N′-Bis(2,6-diisopropyl phenol)–3,4,9,10-perylenetetracarboxylic Diimide (PDI-O, Supplementary Fig. 1) was selected to act as the model compound to achieve room temperature polariton condensation for the following two reasons: (1) PDI-O could provide Frenkel excitons with large transition dipole moment and high binding energy owing to the planar, rigid, and extensively π-conjugated backbone[33], which is favorable for room temperature strong exciton–photon coupling (Supplementary Figs. 2 and 3); (2) the 2,6-diisopropylphenyl substituents in the imide position of PDI-O (Fig. 1a) can increase face-to-face intermolecular distance and therefore prevent π–π interactions between adjacent molecules[34], which would otherwise cause non-radiative decay that hinders polariton condensation in crystalline structure[35].

Here, PDI-O microcrystals were fabricated through a liquid-phase self-assembly method (Supplementary Fig. 4). As shown in the scanning electron microscopy (SEM, Fig. 1b) and atomic force microscopy (AFM, Supplementary Fig. 5) images, the as-prepared PDI-O microcrystals have a well-defined ribbon-like morphology with smooth surfaces and flat side facets, which enables the microcrystals to function as lateral waveguide F–P cavities for strong coupling. The cavity direction is parallel to the ribbon width, and the $Q$ factor of the cavity was estimated to be ranged from 130 to 170 (Supplementary Fig. 6), corresponding to a photon lifetime of ~50 fs. The X-ray diffraction (XRD, Supplementary Fig. 7) pattern indicates that the highly crystalline microribbons have a monoclinic crystal structure belonging to P121/n1 space group. In this crystal structure, the J-aggregation of PDI-O molecules (Supplementary Fig. 8) would reduce non-radiative decay caused by π–π interactions, which helps to provide high-density Frenkel excitons to strongly couple with microcavity photons[36]. Analyses on the transmission electron microscopy (TEM) and selected area electron diffraction (SAED) patterns (Fig. 1c) reveal that the PDI-O microribbon grows along the [010] direction, which suggests that the transition dipole moment of excitons is perpendicular to the ribbon width (Fig. 1d and Supplementary Fig. 9). Because the light–matter coupling strength is described by the scalar product of transition dipole moment and the cavity electric field[31], such orientation would be beneficial for strong coupling with maximized coupling strength (Supplementary Fig. 10)[37–39], which is important for BEC considering the relatively low $Q$ factor of the microribbon cavities. Under optical pumping, the strong coupling between Frenkel excitons and photons generates polaritons in the microribbons, which might relax to the bottom of the lower polariton branch (LPB) by stimulated scattering, possibly forming polariton Bose–Einstein condensates (Fig. 1e).

Angle-resolved microphotoluminescence (AR μ-PL) measurements were carried out to investigate strong coupling in the PDI-O microribbons (Fig. 2a). As illustrated in Fig. 2b, the guided light travels along $x$-direction in the microribbon cavity and diffracts out at the lateral facets, forming point-like sources that emit photons nondirectionally in the $x$–$z$ plane[40]. The emitted photons were detected in the $y$–$z$ plane at different angles, which reflect the polariton dispersion because the wave vector component along $y$-axis ($\mathbf{k}_y$) depends upon the angle of propagation ($\theta$). Figure 2c presents the AR μ-PL spectrum of a PDI-O microribbon excited with a 405 nm continuous-wave (CW) laser. The microribbon supports multiple waveguide cavity modes, which can simultaneously couple to exciton resonance of PDI-O, resulting in several polariton branches that show apparently

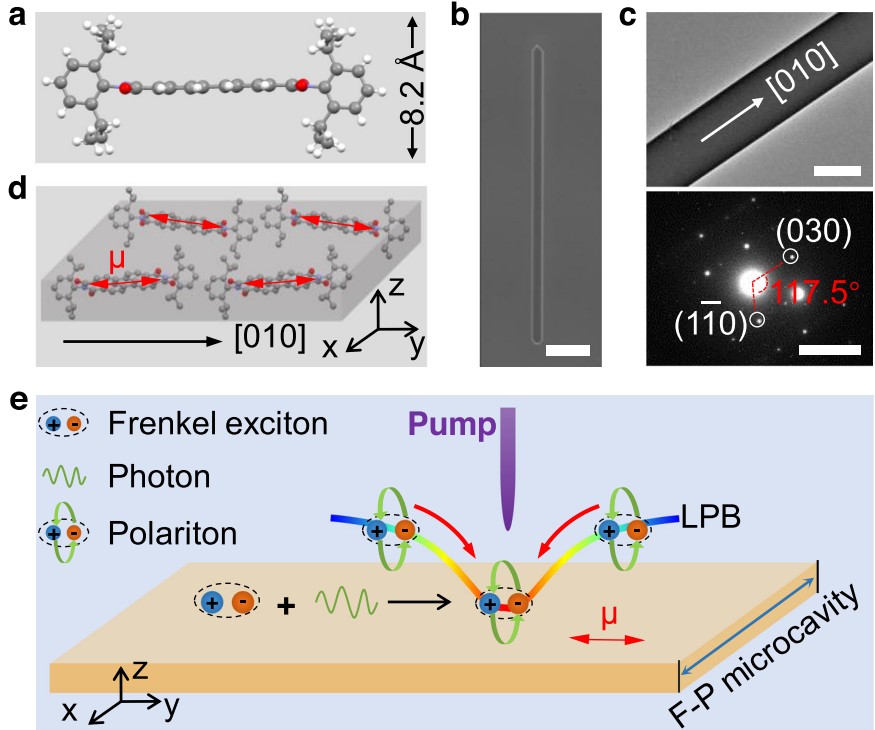

**Fig. 1 PDI-O single-crystal microribbon cavities for polariton Bose–Einstein condensation. a** Chemical structure of the PDI-O molecule. The 2,6-diisopropyl phenol groups in imide position result in a large distance between π-planes (4.1 Å), preventing π–π interactions in crystalline state. **b** SEM image of a PDI-O microribbon. Scale bar: 25 μm. **c** TEM image and corresponding SAED patterns of a PDI-O microribbon. Scale bars are 5 μm and 2 1/nm, respectively. **d** Spatial relationship between the PDI-O molecular transition dipole moment (red double-headed arrow) and the [010] growth direction (black arrow) of the PDI-O microribbon. The dipole direction is along the ribbon length, which is defined as y-axis. **e** Schematic illustration of polariton condensation in an organic microcrystal cavity. LPB: lower polariton branch.

dependence on the microribbons width and thickness (Supplementary Fig. 11). The dispersion curvatures become smaller at short wavelengths and show a repulsion-like behavior at large angles, and such anti-crossing behavior, albeit not much evident due to the large exciton-cavity detuning, indicates unambiguously the occurrence of strong coupling in the PDI-O microribbon.

To gain deeper insight into the strong coupling behavior, we performed a theoretical fitting to the measured polariton dispersion using the coupled oscillator model (Supplementary Note 1)[41]. The results show a good fit between the calculated polariton dispersion curves and the experimental AR μ-PL spectrum (Fig. 2c), confirming that the measured PL is the result of polariton emission. The polariton emission is located near 580 nm, which corresponds to the energetic separation between exciton reservoir ($S_{10}$) and the first vibronic sublevel of the molecular ground state ($S_{01}$), indicating molecular vibration-assisted population of polaritons from the exciton reservoir (Supplementary Fig. 12)[42]. Rabi splitting of $\Omega = 530$ meV extracted from the fitted data indicates a very large coupling strength, which is resulted from the high molecular density in the crystal, the large transition dipole moment of PDI-O molecule, and the maximum overlap between the transition dipole moment and the electric field in the microcavity[31,32,35,38]. The PL spectrum of the PDI-O microribbon (Fig. 2d) shows clear multiple resonance peaks with unequal mode spacing, which correspond to the eigenmodes of the exciton polaritons. By measuring the mode spacing and using the coupled oscillator model with fixed fitting parameters, we calculated the refractive index ($n$) of the microribbon (see Supplementary Note 2) and plotted $n$ as a function of wavelength in Fig. 2e. The calculated refractive index is approximately equal to the background refractive index (Supplementary Fig. 13) at longer wavelength

and shows a remarkable increase at wavelengths close to the excitonic resonance of PDI-O (525 nm). This anomalous dispersion of refractive index verifies the existence of exciton–polaritons in the microribbons[43], which is a prerequisite for the realization of polariton condensation.

**Bose–Einstein condensation of exciton polaritons**. Polariton Bose–Einstein condensation in the PDI-O microribbon cavity was investigated using a pulsed pump laser (400 nm, 150 fs, 1 kHz). At low pump fluence (1.8 μJ cm$^{-2}$), PL spectrum shows emission of multiple polariton modes (Fig. 3a), indicating that these polariton branches are all populated. With pump fluence increasing, the emission of one polariton mode at $\theta = 0$ becomes predominant (Fig. 3b) and exhibits a sharp increase of intensity (Fig. 3c), manifesting enhanced macroscopic ground-state occupation—one of typical features for Bose–Einstein condensation. Condensation preferentially occurs at this mode due to efficient molecular vibration-assisted population process[22]. The ground-state emission ($\theta = 0$) intensity exhibits a nonlinear increase at a critical threshold of $P_{th} = 2.3$ μJ cm$^{-2}$ (Supplementary Fig. 14), accompanied by a sharp decrease of the FWHM (Fig. 3d), which indicates the increase in temporal coherence, further evidencing the onset of polariton condensation through stimulated scattering. The scattering mechanism is verified by time-resolved PL measurement results in Fig. 3e, which show that the emission lifetime decreases from 277 to 47 ps when pump fluence is increased above the threshold. Such lifetime is much shorter than that of PDI molecules undergoing stimulated emission[44], indicating a transition from exciton reservoir dynamics to an ultrafast decay process corresponding to the stimulated scattering from the exciton reservoir to the condensate. The condensates feature an

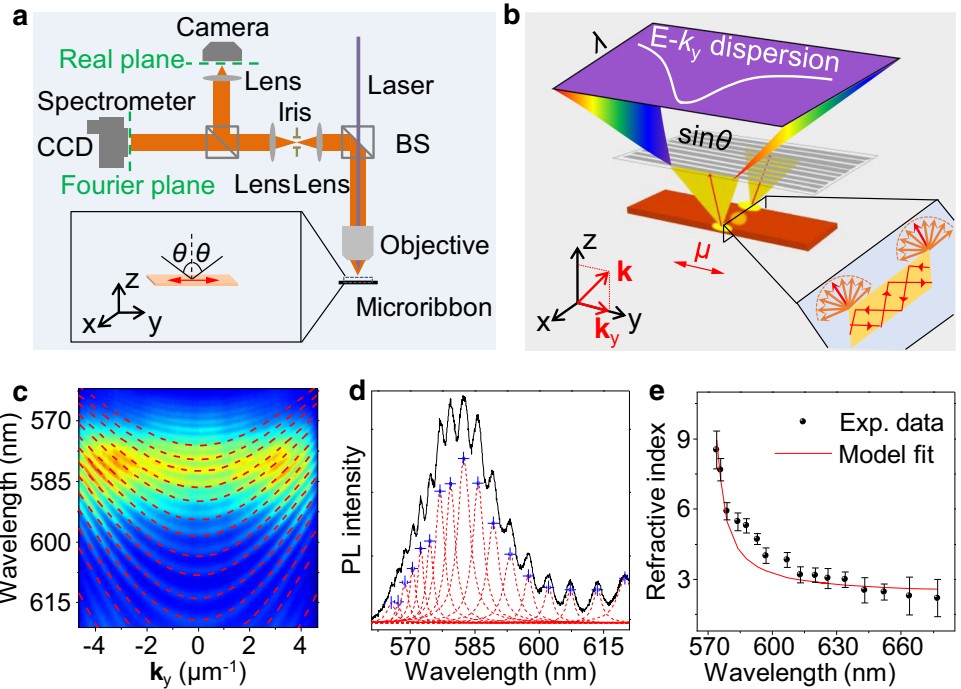

**Fig. 2 Strong coupling in the PDI-O microribbon cavity. a** Schematic of the experimental setup for AR μ-PL measurement. CCD: charge coupled device. BS: beam splitter. **b** Schematic of a PDI-O microribbon along with coordinate axes. The waveguide F–P cavity direction is parallel to the $x$-axis direction. $k_y$ is the projection of wave vector on to $y$-axis, which depends upon the emission angle in $y$–$z$ plane ($\theta$). **c** AR μ-PL spectrum of the PDI-O microribbon. Dashed curves are fitted dispersions of polariton modes obtained from the coupled oscillator model. The microribbon was excited by a 405 nm CW laser. **d** PL spectrum of the PDI-O microribbon. The dashed lines are fitted Lorentzian line shapes to measure the mode spacing $\Delta\lambda$ for calculation of refractive index. Error bars indicate the 95% confidence intervals of the fitted peak positions. **e** Refractive index derived from the coupled oscillator model (red line) and calculated from experimental data (black dot). Error bars indicate the 95% confidence intervals propagated from the 95% confidence intervals for the fitting of the PL spectrum in **d**.

emergence of long-range spatial coherence, which was probed by Young's double-slit interferometry experiment (Supplementary Fig. 15)[45]. The recorded pattern shows barely visible interference fringes below threshold (Fig. 3f), while above threshold distinct interference fringes are clearly observed with the visibility contrast of 35% (Fig. 3g), indicating the emergence of spatial coherence of BEC in the microribbon. Accordingly, owing to the effective scattering mechanism that is believed to result from the large transition dipole moment[46], polariton Bose–Einstein condensation in the PDI-O microribbon was achieved without the requirement of external cavity.

**Manipulation of polariton condensates for controllable coherent light output**. In polariton Bose–Einstein condensates, the polaritons decay by the leakage of photons from the microcavity, which enables the PDI-O microribbons to function as coherent light sources[47]. When the threshold was reached, the formed polariton condensates are localized and have no lateral wave vector component (Fig. 4a), and therefore the coherent light emission occurred at the pump area, with an emission angle (defined as angle between emission direction and $z$-axis in $y$–$z$ plane) of zero[48]. It has been reported that when the pump power is high enough, the interactions between polaritons and the exciton reservoir can drive polaritons to propagate over macroscopic distances[49–51], which may provide us an approach to manipulating polariton condensates for controllable output of coherent light. As illustrated in Fig. 4b, the photo-generated high-density exciton cloud in the excitation area could interact with polaritons, producing a force that expels polaritons from where they are formed. Due to the dimensional confinement of the microribbon, polariton condensates might be driven to propagate

along the microribbon and acquire a finite wave vector in the direction away from the excitation area. Considering the high structural homogeneity of the single-crystal microribbons[52,53], polaritons could travel over macroscopic distances within their lifetime which was estimated to be ~50 fs. As a result, coherent light would be emitted at positions separated from the excitation area, with emission direction deviating toward the side opposite to the excitation point.

Exactly, this is what we have observed from the experiment. With increasing pump power, the AR μ-PL spectra in Fig. 4c exhibit a power-dependent increase of emission angle, indicating that the polaritons have obtained a wave vector along with the ribbon, which is resulted from the repulsion of excitons in the excitation area (Supplementary Fig. 16)[49,54]. Meanwhile, the PL images show strong emission outside the pump area at high pump power, which indicates that the polaritons have propagated outward until decayed into photons (Supplementary Fig. 17)[55]. The right panel of Fig. 4c plots the horizontal cross-section intensity profiles of the PL images (black), which agree well with the wave function probability density (red) calculated by solving Gross–Pitaevskii equation, where a pump power-dependent repulsive potential is taken into account, further evidencing the polariton–exciton repulsive interactions in the condensates (Supplementary Note 3). As the repulsive potential is proportional to exciton densities that are determined by temperature-dependent decay processes[56–58], we were able to further control the emission angle by changing temperature (Fig. 4d). The result not only confirms the existence of polariton–exciton interactions in the PDI-O microribbons, but also presents the potential for the construction of miniaturized coherent light-emitting devices with multiple controllability.

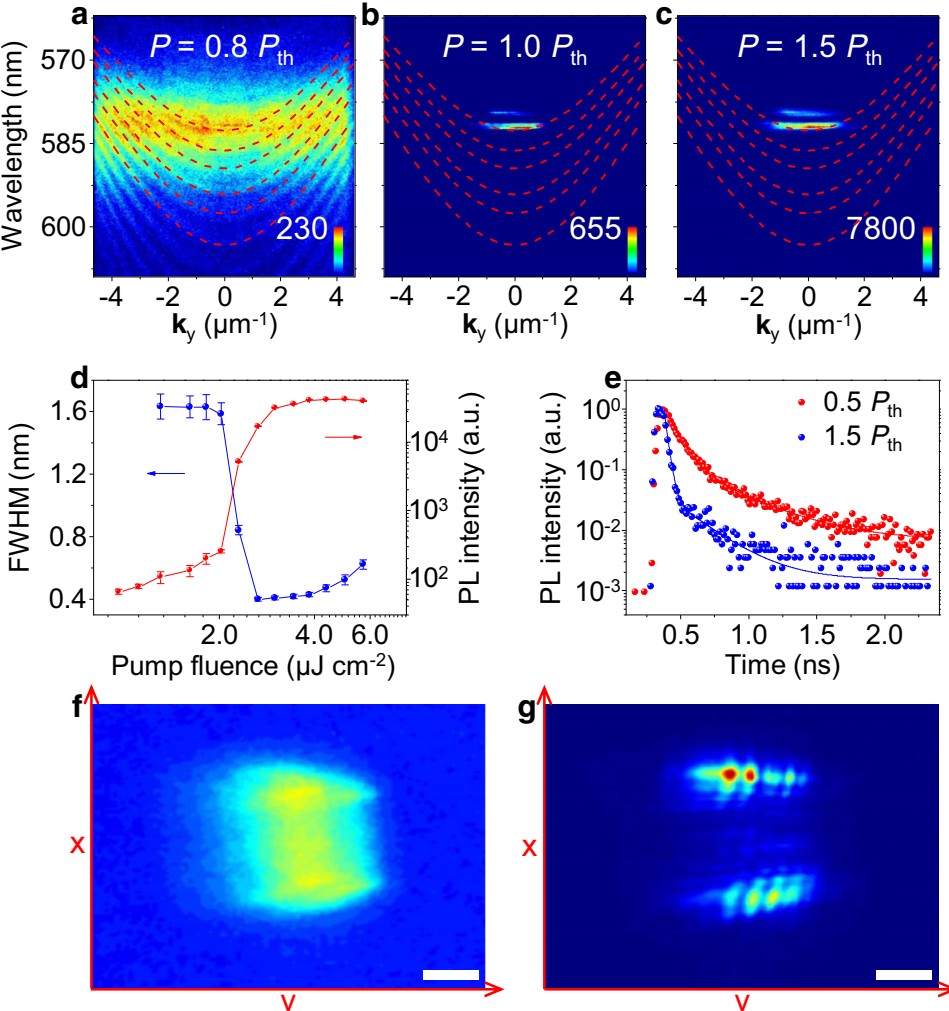

**Fig. 3 Polariton condensation in the PDI-O microribbon. a–c** AR μ-PL spectra of the PDI-O microribbon measured at $0.8P_{th}$, $1.0P_{th}$, and $1.5P_{th}$, respectively. Dashed curves are fitted polariton dispersion. When the pump fluence reaches a threshold, the polariton emission collapses to the bottom of the polariton dispersion with a sharp increase of intensity, suggesting the onset of polariton condensation. The microribbon was pumped by a pulsed laser (400 nm, 150 fs, 1 kHz). **d** Emission intensity and FWHM at $k_y = 0$ ($\theta = 0$) as a function of pump fluence. The PL intensity was plotted on a log–log scale. Error bars indicate the 95% confidence intervals obtained from the fits to the measured PL spectra. **e** The time-resolved PL decay profiles at $0.5P_{th}$ (red) and $1.5P_{th}$ (blue). The lifetimes below and above $P_{th}$ are 277 and 47 ps, respectively. **f** The image of the interference pattern in Young's double-slit interferometry experiment below the threshold. **g** The image of the same microribbon recorded above the threshold. Scale bars are 8 μm. The x- and y-direction in **f** and **g** correspond to the direction of the microribbon width and length, respectively.

In conclusion, we have reported room-temperature polariton Bose–Einstein condensation in organic PDI-O single-crystal microribbon cavities. In these microribbon F–P cavities, the excitons of PDI-O molecules readily undergo strong coupling with microcavity photons at room temperature owing to their Frenkel nature and large transition dipole moments. The high exciton densities in the single crystal leads to a very large coupling strength despite the low cavity Q, which facilitates Bose–Einstein condensation of the formed polaritons in the microribbon cavities. Based on the coherent nature and manipulability of polariton condensates, we further demonstrate controllable coherent light output from the PDI-O microribbons by virtue of the repulsive interactions between polaritons and the exciton reservoir. Our results provide inspiration for the potential of organic material for constructing miniaturized polaritonic devices towards practical applications in photonic circuits.

## Methods

**Materials**. 3,4,9,10-Perylenetetracarboxylic dianhydride (98%), 2,6-diisopropyla-niline (95%) and imidazole (99%) were purchased from InnoChem (Beijing, China), which were used to synthesize PDI-O according to the literature procedure[59].

**Preparation of PDI-O microribbons**. The PDI-O microribbons were prepared through a solvent-evaporation-induced self-assembly method. In a typical preparation, 100 μL of PDI-O dichloromethane solution (1 mM) were dropped onto a quartz substrate, which was placed in a beaker (20 mL) containing 5 mL dichloromethane. In the whole procedure, a solvent atmosphere was created and the evaporation speed of the solvent (i.e. the saturation degree of the solution) can be well controlled. With self-assembly of the PDI-O in the beaker for 2–3 days, large area microribbons were finally obtained.

**Quantum chemical calculation**. The ground-state geometry of PDI molecule was optimized at the B3LYP/6–311G* level with the Gaussian 09 package. Based on the optimized geometry, the transition dipole moment vector (**μ**) of PDI from $S_0$ to $S_1$ was theoretically calculated by Multifwn program at the same level.

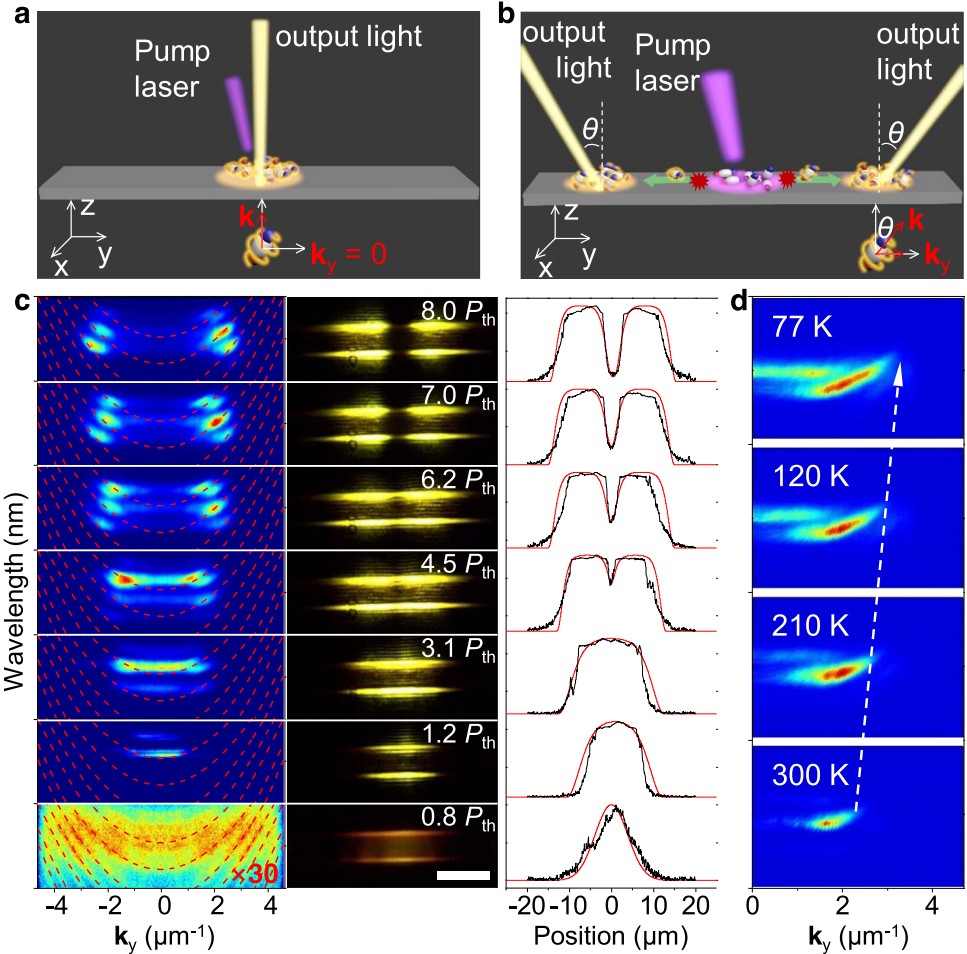

**Fig. 4 Controllable coherent light output based on repulsive interactions within polariton condensates in a PDI-O microribbon. a** Schematic illustration of light output from EP BEC at low pump power. The polariton condensates decay into coherent photons, producing light from the excitation area. As polaritons have no wave vector along the microribbon ($\mathbf{k}_y = 0$), the detected output direction is along z-axis. **b** Schematic illustration of light output from EP BEC at high pump power. The repulsive interactions between polaritons and the photo-generated excitons drive polaritons to propagate along the microribbon. The polariton condensates flow outward the excitation area and acquire a finite wave vector ($\mathbf{k}_y$), leading to light emissions at nonzero angles. **c** Left: angle-resolved PL spectra at different pump powers. Middle: the corresponding PL images of PDI-O microribbons. Scale bar is 10 μm. Right: horizontal cross-section intensity profiles of the PL images (black) and the wave function probability density calculated by solving Gross–Pitaevskii equation (red). **d** Temperature-dependent AR μ-PL spectra of a PDI-O microribbon measured at same pump fluence (7 μJ cm$^{-2}$). The emission angles become larger with temperature decreases (white arrow).

**Finite-difference time-domain (FDTD) simulation**. The electric field intensity distribution in a single microribbon was calculated by FDTD simulation (FDTD solutions, Lumerical Solutions, Inc). The model consists of a microribbon (15 μm × 2 μm × 200 nm) with a dipole source located at the center. The dipole source is parallel to the ribbon length, and the simulation domain is surrounded by perfectly matched layers. The background refractive index of the microribbon was set to be $2.2 + 0.01i$, with the imaginary part introducing a loss resulted from re-absorption of the PDI-O molecules. The refractive index at the center area is slightly higher than the background value (2.3), and the imaginary part was removed considering that light is generated and amplified upon laser pumping. Because the microribbon is excited with a dipole source, there exist short-lived transients resulted from the strong emission of the dipole near the start of the simulation. To filter away these unwanted transients and get the correct mode profile for the cavity, we used a time apodization (start apodization, center: 20 fs, width: 2 fs) in the Monitor.

**Morphology and structural characterizations**. The morphology of the PDI-O microribbons was characterized by SEM (FEI Nova NanoSEM 450), TEM (JEOL JEM-1011), and AFM (Bruker Multimode 8). The crystal structure of the PDI-O microribbons was examined by SAED (JEOL JEM-1011) and XRD (PANalytical Empyrean).

**Optical characterization**. The absorption spectra of the PDI-O microribbons were measured on a UV–visible spectrophotometer (Hitachi, UH4150). The absolute quantum yields were measured by using the Hamamatsu Absolute Quantum Yield

Spectrometer C11347. Angle-resolved photoluminescence spectra were measured by AR μ-PL setup based on Fourier optics. The excitation pulses (400 nm) were generated by the second harmonic of the fundamental output of a regenerative amplifier (Spectra Physics, 800 nm, 100 fs, 1 kHz), which was in turn seeded by a mode-locked Ti:sapphire laser (Mai Tai, Spectra Physics, 800 nm, 100 fs, 80 MHz). The emission of the PDI-O microribbons was recorded using a spectrometer (Princeton Instruments, IsoPlane SCT 320) equipped with a thermal-electrically cooled CCD (Princeton Instruments, PIX-1024BX). The time-resolved photo-luminescence of the PDI-O microribbons were measured by a streak camera (Hamamatsu photonics, C10910).

The AR-μ-PL spectra were measured through a setup with Fourier optics. The photoluminescence of the PDI-O microribbon is focused at the rear focal plane of the microscope objective (×100, numerical aperture, NA = 0.8), which is then projected onto the entrance slit of a spectrometer equipped with a CCD. Therefore, the two orthogonal axes of the CCD correspond to the wavelength λ (equivalently as energy, E) and angle θ (equivalently as wave vector, $\mathbf{k}_y$) of the emitted light, respectively. When the long axis y of the PDI-O microribbon is set to be parallel to the entrance slit of the spectrometer, we can get the dispersion of the polaritons in the ($\mathbf{k}_y$, λ) plane.

## Data availability

The data that support the findings of this study are available from Y.S.Z. upon reasonable request.

## Code availability

The Finite-difference time-domain source code can be accessed from http://www.fdtdxx.com.

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

## Acknowledgements

This work was supported financially by the Ministry of Science and Technology of China (No. 2017YFA0204502) and the National Natural Science Foundation of China (Nos. 22090023, 21790364 and 91750103).

## Author contributions

Y.S.Z. conceived the original concept and supervised the project. J.T. and H.W. designed and performed the experiments and prepared the materials. J.Z., L.S., and J.T. performed the theoretical calculations. J.T., F.F.X., Y.L., C.Z., J.Y., and Y.S.Z. analyzed the data and wrote the paper. All authors discussed the results and commented on the manuscript.

## Competing interests

The authors declare no competing interests.
