## [Peer Review File · Nature Communications]

REVIEWER COMMENTS

Reviewer #1 (Remarks to the Author):

In this manuscript, authors demonstrate room temperature BEC in organic single crystal microbelt cavities. The work is exciting and will be of interest to community in strong coupling, exciton-polaritons and organic semiconductor lasers and optoelectronics at large. As such, the manuscript is appropriate for Nature Communications, however, in its current form can't be accepted for publication. The manuscript is confusing in its explanation of polariton condensation and emission. While most claims are supported by experiments, polariton features such as anticrossing behaviour, direction of emission, polariton repulsion don't appear to be clearly explained and/or demonstrated - specific comments below. I recommend authors to address these comments in a revised version of their manuscript for further consideration.

1. Authors mention they are optical pumping from the top (z axis) but the cavity direction is actually the width (which is $\sim 5 \mu\text{m}$). Why can't all be done through the top and bottom facets? Is the Q-factor too low? What is the height of microbelts?
2. On a similar note as 1, all emission is collected from z-axis, Fig 2. Why aren't the polariton features along the direction of the FP cavity? Emission from point dipoles forms a cone so it's understandable some emission to be seen at large angles but it's not clear why no emission is observed at 0 degrees (i.e. 90 degrees away from direction of FP). Also, the laser emission comes out at 0 degrees, Fig 3. How does that work?
3. Authors say "As polaritons have no wave vector along the microbelt ($k_y = 0$), the output light direction is along z-axis." It's not clear why is that. Why would emission not appear in the direction of the FP? It's clear from Figure S12 that the number of modes are changing with the width of the microbelt, so it seems like this is emission coming from FP resonance peak (FP from z direction would have much bigger spacing between modes).
4. Authors should discuss why anticrossing behaviour is not evident from PL spectra.
5. It's unclear how authors support the claim that the 8x threshold spectrum is from repulsive behavior of polaritons. Usually, the repulsive behavior just results in higher energy emission. Is it higher order mode lasing?
6. In SI Figure 7, text says: "The diffraction peaks in the XRD pattern of the PDI-O microbelts well match those of the simulated results from the single-crystal diffraction of PDI-O", but the two XRD spectra don't look alike at all. What is been compared here?
7. Have authors tried approaching the threshold for regular photonic lasing? It would be interesting to see if they can observe both thresholds or if the material degrades first.
8. Why does the electric profile of Suppl Fig 10 (left image) look the way it does, i.e. why strong signal are only observed near the edges? Reconcile them with points 1 and 2.
9. How do these results compare to recent cavity free polariton work on crystals from uniaxially aligned oscillators by Meskers, topology work from Yuen-Zhou, single crystal microcavity work including J-aggregates from Bittner, Silva, Spano and others? Authors should consider placing their work in a wider context for readers to gain a deeper appreciation for progress in this field.

Reviewer #2 (Remarks to the Author):

The authors report on the realization of a polariton condensate at room temperature in so-called microbelt cavities, microcrystals, which form during the self-assembly process and may serve as low-Q microcavities. They find that in these microbelt cavities polariton condensation indeed becomes possible under pulsed excitation. The polariton dipole moment is found to show a strong directionality, which results in a significant dependence of the emission on the polarization and geometric orientation. The authors also find that the momentum of the polaritons depends strongly on the pump power.

With respect to the validity of the results here, I tend to believe the authors' claim that polariton condensation is actually taking place. The authors show only limited experimental data - including the line width, output intensity and decay time of the mode. However, the momentum redistribution discussed at the end of the manuscript will usually occur only in the condensed regime, which is somewhat reassuring. However, I strongly suggest that the authors either should add more experimental points to the input-output curve shown in figure 3)d) or (if higher cw powers should break the system) instead show an input-output curve for pulsed excitation as well. At current, the input-output curve is not convincing at all. I further suggest strongly that the authors show more detailed data on the momentum redistribution for pulsed pumping for different pump powers. If the polariton redistribution in momentum space shows a reasonable dependence on pump power, I would take this as a reasonable experimental signature of condensation taking place. In order to establish the material platform in the long run, it will also be necessary to investigate spatial coherence and second-order coherence, but these two points be a topic for future manuscripts. With respect to the potential impact of the present manuscript, polariton condensation at low quality factors is certainly of interest to the wider community, especially as it is very difficult to achieve large quality factors for organic materials.

In addition to the points just mentioned, although the manuscript certainly provides some interesting aspects, I cannot recommend it for publication in the present form due to some shortcomings that in my opinion put the manuscript below the standards of Nature Communications at current. However, it might be possible to overcome these problems by further changes additions to the manuscript. There are some minor points to be listed later on, but my main reservations with respect to the validity and the impact of the results presented here are twofold. First, for some aspects there is very little quantitative information given. Accordingly, it will be hard for the reader to correctly judge and even reproduce the results presented here. Second, there are some inconsistencies in the explanations given here, which definitely must be taken care of. Let me express my concerns in detail:

1) One of the main selling points of the authors consists of the fact that they are able to realize polariton condensation even in low-quality organic cavities due to the rather large dipole moments they are able to achieve. However, the author never gets to know what a low cavity quality factor actually means in this case. As the cavities used in organic polaritonics usually show low Q factors as compared to epitaxially grown semiconductor microcavities anyway, the reader will need some quantitative information to be able to judge how much of an advancement the results presented here actually are. I am aware that there are plenty of cavity modes supported by the structure used here and measuring the Q factor exactly is a nontrivial task, but this task is indeed easier for low-Q modes. Accordingly the number should be presented. This number will in turn be required, so the reader can judge the importance of the other main result of the authors correctly: Control over the polaritons via the interaction with the exciton reservoir. Indeed, this is an interesting phenomenon, but it is usually considered to be most useful in the case of large-Q cavities, where the distances traveled by the polaritons within their lifetime may become mesoscopic. As the cavity quality factor also determines the polariton lifetime, it is in my opinion necessary to provide the polariton lifetime in the present structure as well, so the reader may judge whether the travel distances that may be achieved are actually long enough for "constructing miniaturized polaritonic devices towards practical applications in photonic circuits" or not.

2) Maybe some more critical points are given by inconsistencies within the manuscript. The most important problem given in the manuscript is the fact that the authors never clearly define which cavity mode they are actually interested in. Most of the experimental studies presented here focus

on an experimental geometry, where angle-resolved (and therefore momentum-resolved) spectra are shown and the relevant angle is the one between the z-axis and the y-axis, where the angle is chosen such that the zero-angle corresponds to an orientation along the z-axis and the $\sin(\theta)$ values shown e.g. in figures 3 and 4 correspond to the wavevector projection k_y along the y-axis. The authors find that the dispersion shows a minimum for $k_y=0$ and shows a parabolic dispersion for low values of k_y . This is the typical and expected result for a cavity mode, where photons are confined along the z-direction, which will typically be called the cavity direction. Instead, in calculations, e.g. in the supplementary figure 10 in the supplementary material, the authors discuss several different orientations of the dipole within the x-y plane and discuss that this change corresponds to different angles between the dipole and the cavity direction. This means that the cavity direction is now aligned along the x-axis, which would be in line with the depiction given in figure 1)e). This is obviously inconsistent. To complicate matters further, the authors frequently refer to the width and length of the cavity as well as to its short and long axis. The directions of those are never explicitly defined within the manuscript, which will make it impossible for the reader to identify the geometry at hand. The same goes for the comparison between TE and TM modes in the supplementary material, which is a great supporting information, but lacks the information with respect to which axis of the structure TE and TM are defined (or equivalently: which cavity mode is considered). A thorough introduction of the exact geometry used will be highly beneficial for the reader.

3) A minor point, which is still relevant: The strong directionality of the dipole moment within the microbelt structures effectively renders them microwire-like effective 1d structures. On the page showing figure 12 in the supplementary material, the authors discuss the appearance of several modes with unequal mode spacings. The expected mode spacings for 1d microwires of a given width are well-known. Considering the confusion about the experimental geometry already mentioned, it would be very helpful to show that the bare cavity mode energies, which should be obtainable from the coupled oscillator model already applied by the authors, follow the expected trend for the given microbelt widths. This might be important as some of the cavity modes might not couple efficiently to the excitons due to the strong orientation of their dipole moments. Further, it is known that the polariton spectrum may become quite non-trivial including complicated energy shifts in the presence of relaxation processes (Phys. Rev. B 82, 245315 (2010)), so the authors' claims would become much more substantial if they could provide a clear explanation for the mode spectrum in their structure. Along the same lines, the authors might also want to add an explanation on why polariton condensation does not occur in the polariton mode arising from the lowest energy cavity mode.

With respect to the state of the art in the field, I am aware that the field is large and it is impossible to cite every single paper that is relevant with respect to the topic at hand, but in my opinion there are some shortcomings with respect to adequately referencing the work of others working in the field:

The authors utilize the idea that repulsive interactions between polaritons and excitons and polaritons result in a controllable polariton emission pattern in momentum space. The explanation for this effect was presented first in an important paper by Wouters et al. (Phys. Rev. B 77, 115340 (2008)), which definitely deserves to be cited in the present manuscript. Further, as mentioned above, Phys. Rev. B 82, 245315 (2010) might be a relevant paper here.

Some additional minor points with respect to clarity and presentation:

- The dashed lines that are supposed to show the fitted polariton dispersions in figures 3)a) to c) are almost impossible to see. The authors might want to use thicker lines and/or a different color.
- Within the main text, the authors highlight that the main difference between the spectra shown in figure 2 and 3 is the usage of cw versus pulsed excitation. As this is quite important information, it might improve the ease of reading the manuscript if this information was added to the figure captions as well.
- On page 3, the authors judge that "planar microcavities tend to have large footprints". That statement seems too vague to me to make sense for a general readership.
- The authors frequently show angle-resolved emission spectra. It would be helpful for the reader to directly convert the value shown to the wavevector along the y-direction.

In summary, I cannot recommend the current manuscript for publication in Nature

Communications, but a significant revision might be able to meet the journal standards.

Reviewer #3 (Remarks to the Author):

The manuscript by Zhao and co-workers claims observation of room temperature polariton Bose-Einstein condensation in organic single-crystal "microbelt" cavities. Such "microbelts" are formed by the organic compound PDI-O through a self-assembly process, and function as Fabry-Pérot microcavities. The PDI-O molecules are densely packed and arranged in such a way in the microcavities that the structure is capable of supporting large densities of Frenkel excitons with large transition dipole moments. Evidence of strong coupling is obtained through angle-resolved photoluminescence spectroscopy, which allows the authors to extract dispersion curves for the FP modes, and fit with a simple model. Evidence of Bose-Einstein condensation is the observation of spectral narrowing and lifetime decrease for one of the FP cavity modes, as a function of pump fluence. The authors also claim the ability to manipulate the condensate, evidenced in the evolution of the emission spatial profile with pump fluence, which suggests repulsion of excitons in the illumination area.

I find that more clarification is needed for the evidence presented, before I can recommend publication. The following comments must be addressed:

1 - The term microbelt is somewhat misleading, as it evokes a circular closed structure. I suggest calling it something else, like 'nanowaveguide' or something else that reflects the 1D linear geometry.

2 - In the angle-resolved PL maps, I would expect to see discrete islands in the energy X momentum space, corresponding to the different longitudinal FP modes. I.e., for each N, which from eq. (4) corresponds to the transverse mode order, rather than a continuous branch as a function of θ , I would expect to see islands. Why is it not the case, is it an artifact of the plot?

3 - It is not clear where the supplementary expressions (2) & (3) for the FP cavity mode dispersion came from. This needs to be clarified, since a fit to the angle-resolved PL with this model is used to justify strong coupling. It would probably be very helpful also if the authors could present the dispersion curves for the empty cavity, to make the "repulsion-like" behavior, which can be a signature of strong coupling, more obvious.

4 - The intensity of the PL peaks decreases considerably at higher energies, where the large refractive index increase is observed. How well are the FP peak positions known, in this case? For instance, if a fit was used, error bars indicating the 95 % confidence intervals for the expected peak positions should be included in the graph.

5- Eq. (4) in the supplement is a measure of the waveguide group index, which is not a function of the material and the waveguide. Waveguide dispersion alone can lead to large group indices. Can the authors calculate the expected group index due to just waveguiding, considering an FP resonator (or waveguide) with the background index, or a proper refractive index extracted at long wavelengths (i.e., far from the exciton)?

6 - The linewidth narrowing observed in Fig. 3d is small. Can the authors report the uncertainties associated with the reported linewidths? I also suggest plotting the PL intensity in log scale, so readers can appreciate the degree of enhancement.

7 - Still regarding the results in Fig. 3, more convincing evidence of BEC formation would be through interferometric measurements of the spatial and temporal coherence. Have the authors tried to perform such measurements

8 - Regarding results in Fig. 3e: reduction of lifetime can be observed in ASE in laser cavities with emitter ensembles, in going from uncorrelated, individual emission to collective emission. Can the authors provide any evidence the observed lifetime reduction is due to stimulated scattering, and not due to the mentioned mechanism?

9 - The explanation for Fig.4, regarding the excitonic repulsion, is very qualitative and should be better supported. Have the authors attempted to simulate the behavior of Fig. 4 with the Gross-Pitaevskii model shown in the supplement ? Doing this could provide better evidence of the achievement of BEC and of the exciton repulsion as claimed.

Response to Reviewer 1

General Evaluation: *In this manuscript, authors demonstrate room temperature BEC in organic single crystal microbelt cavities. The work is exciting and will be of interest to community in strong coupling, exciton-polaritons and organic semiconductor lasers and optoelectronics at large. As such, the manuscript is appropriate for Nature Communications, however, in its current form can't be accepted for publication. The manuscript is confusing in its explanation of polariton condensation and emission. While most claims are supported by experiments, polariton features such as anticrossing behaviour, direction of emission, polariton repulsion don't appear to be clearly explained and/or demonstrated - specific comments below. I recommend authors to address these comments in a revised version of their manuscript for further consideration.*

Response: We thank the reviewer for the nice summary and positive evaluation of our work. We are also very grateful for the insightful comments and constructive suggestions, which have helped us to make a further improvement of our work. In the following, we provide concrete responses to the comments and suggestions point-by-point.

Comment 1: *Authors mention they are optical pumping from the top (z axis) but the cavity direction is actually the width (which is ~5 um). Why can't all be done through the top and bottom facets? Is the Q-factor too low? What is the height of microbelts?*

Response 1: Thanks a lot for the insightful comment. As the reviewer mentioned, the microbelts were optically excited from the top, while the detected angle-resolved PL spectra agreed with the waveguide cavity model where the cavity direction is parallel to the belt width. This waveguide cavity effect involves the travelling of light in the microbelts along the width direction by total internal reflection (*Nano. Lett.* **14**, 6564 (2014)). The atomic force microscopy (AFM) measurement results reveal that the height of the microbelts is typically ~150 nm (Supplementary Figure 5), which is not

sufficient to allow the long range travelling of light in the direction normal to the top facet of the microbelts (z-direction). In addition, the Q factor of z-direction cavity is mostly determined by the reflectance of the top and bottom facets. Thus, we have measured the reflectivity spectrum of a microbelt at normal incidence (Figure R1), which shows a relatively low reflectance (25%) for the top and bottom facets of the microbelt, which is possibly due to the leakage of light into the dielectric substrate. As a result, optical cavity along the belt width is superior to that in z direction, and dominates the spectral modulation that we have observed in this work.

To provide more informative description of the waveguide-involved cavity effect in the microbelt, we have added the word “waveguide” in front of the terms “cavity” or “F-P cavity” in the revised Manuscript and Supplementary Information. We have also added the related discussion of the height of the microbelts in the revised Supplementary Information.

Figure R1 Reflectivity spectrum at normal incidence of a PDI-O microbelt.

Comment 2: *On a similar note as 1, all emission is collected from z-axis, Fig 2. Why aren't the polariton features along the direction of the FP cavity? Emission from point dipoles forms a cone so it's understandable some emission to be seen at large angles but it's not clear why no emission is observed at 0 degrees (i.e. 90 degrees away from direction of FP). Also, the laser emission comes out at 0 degrees, Fig 3. How does that work?*

Response 2: Thanks for the comment which makes us aware of the possible misunderstanding caused by the description of the measurement geometry. As

illustrated in Figure R2, in the microbelt the guided light propagates via total internal reflection and reflects back and forth in the x-direction. These waveguide cavity modes diffract out of the microbelt at the lateral facets, forming point-like sources that emit photons nondirectionally in the x-z plane (*Nano Lett.* **6**, 2707 (2006)). As a result, we were able to collect the cavity mode emission from the z-direction through an objective. The diffraction only occurs at x-z plane due to the small height of the microbelt, and therefore the wave vector in y-axis, k_y (dependent on angle θ), is not affected, which allows us to obtain E- k_y relationship of the waveguide cavity by directly measuring angle resolved PL at y-z plane. When BEC occurs, the polaritons massively occupy the ground state which is at the bottom of the dispersion curve ($k_y = 0$), and therefore the emission only comes at $\theta = 0$.

Figure R2 Schematic of the measurement of dispersion of a PDI-O microbelt waveguide cavity. The direction of the waveguide F-P cavity is parallel to the x-direction. k_y is the projection of wave vector onto y-axis, which depends upon the emission angle in y-z plane (θ).

We have replaced Figure 2a with Figure R2 which would be more informative about the measurement geometry and modified the related description (Page 7, line 4) in the revised Manuscript as “As illustrated in Fig. 2a, the guided light travels along x-direction in the microribbon cavity and diffracts out at the lateral facets, forming point-like sources that emit photons nondirectionally in the x-z plane³⁹. The emitted photons were detected in y-z plane at different angles, which reflect the polariton dispersion because the wave vector component along y-axis (k_y) depends upon the angle of propagation (θ).”. We have also added related reference (*Nano Lett.* **6**, 2707

(2006)) in the revised Manuscript.

Comment 3: *Authors say “As polaritons have no wave vector along the microbelt ($k_y = 0$), the output light direction is along z-axis.” It’s not clear why is that. Why would emission not appear in the direction of the FP? It’s clear from Figure S12 that the number of modes is changing with the width of the microbelt, so it seems like this is emission coming from FP resonance peak (FP from z direction would have much bigger spacing between modes).*

Response 3: We thank the reviewer for the comment. As is discussed in **Response 2**, the angle-resolved PL spectra we measured from the z-direction represents the dispersion of the waveguide cavity whose direction is parallel to the y-direction (E - k_y relationship). In Figure R2, because the point-like sources at two edges of the microbelt emit photons nondirectionally in the x-z plane, the E - k_y dispersion could also be obtained by collecting the emission from any direction in x-z plane, including the F-P cavity direction (x-direction). The measurement of the emission from the x-direction faces technical difficulties as the microbelt is lying on the substrate due to the small height, and therefore we obtained the dispersions by measuring the angle-resolved PL spectra from the z-direction alternatively.

To avoid confusion, we have modified the related description (Page 24, line 5) as “*As polaritons have no wave vector along the microbelt ($k_y = 0$), the detected output direction is along z-axis.*” in the revised Manuscript.

Comment 4: *Authors should discuss why anticrossing behaviour is not evident from PL spectra.*

Response 4: Thanks for the suggestion which helps us to improve the discussion of the polariton dispersion in our microbelts. In the PDI-O microbelts, because the lower polariton branches (LPBs) are directly populated from the exciton reservoir via vibrationally assisted scattering, the polariton energy is located near 2.14 eV (580 nm) that corresponds to the energetic separation between the exciton reservoir and the first

vibronic sublevel of the molecular ground state (*Adv. Funct. Mater.* **21**, 3691 (2011); *Nat. Photon.* **4**, 371 (2010)). From the absorption spectrum of PDI-O (Supplementary Fig. 3), we can find that the energy of excitons in the microbelts is 2.36 eV (525 nm), which is quite far away from the polariton energy, indicating a giant exciton-cavity detuning. Due to such large energy separation between exciton and cavity modes, the decrease of the curvatures of the mode dispersion is not significant, and therefore the anticrossing behaviour is not evident from the angle-resolved PL spectra.

Following the reviewer's suggestion, we have modified the related discussion (Page 7, line 13) in the revised Manuscript as “*The dispersion curvatures become smaller at short wavelengths and show a repulsion-like behavior at large angles, and such anti-crossing behavior, albeit not much evident due to the large exciton-cavity detuning, indicates unambiguously the occurrence of strong coupling in the PDI-O microbelt.*”.

Comment 5: *It's unclear how authors support the claim that the 8x threshold spectrum is from repulsive behavior of polaritons. Usually, the repulsive behavior just results in higher energy emission. Is it higher order mode lasing?*

Response 5: Thanks for the comment. As has been demonstrated in previous work, the interaction between polaritons and excitons results in the blueshift of the condensate energy (*Nat. Mater.* **13**, 271 (2014)). In addition, if the interaction is spatially anisotropic, it might also lead to the flow of polaritons condensates (*Phys. Rev. B* **77**, 115340 (2008)).

Here, to provide supportive evidence of the repulsive interactions, we have studied the dependence of the polariton momental and spatial distribution on pump power by measuring angle-resolved PL spectra and the corresponding PL images of a microbelt at different pump powers. From the angle-resolved PL spectra (left panel in Figure R3), we can observe a blueshift with increasing pump power, suggesting the repulsive interactions in the condensate. More importantly, the wave vector of the polaritons increases at high pump power, which indicates that polaritons have undergone a

lateral acceleration in the power-dependent potential resulting from repulsive interactions. The PL images (middle panel in Figure R3) show that at high pump power the microbelt exhibits strong emission outside the pump area, indicating that polaritons have propagated along the microbelt until decayed into photons. These results reveal that at high pump power, repulsive interactions between polaritons and the high-density excitons in the excitation area have created a force, and therefore the polaritons are expelled from where they are formed. The right panel of Figure R3 plots the horizontal cross-section intensity profiles of the corresponding PL images (black), as well as the wave function probability density (red) calculated by solving Gross-Pitaevskii equation where a pump power-dependent Gaussian potential resulting from repulsive interactions is taken into account. The great agreement between the experimental and theoretical results further evidence the polariton-exciton repulsive interactions in the condensates.

Figure R3 Left: angle-resolved PL spectra at different pump powers. Middle: the corresponding PL images of PDI-O microbelts. Scale bar is 10 μm . Right: horizontal cross-section intensity profiles of the corresponding PL images (black) and the wave function probability density calculated by solving Gross-Pitaevskii equation (red).

The angle-resolved PL spectra also exhibits emission from different polariton branches at high pump power, indicating the condensation of polaritons with different mode order. This, however, does not contradict the conclusion that the repulsive interactions result in spatial and momental redistribution of the polariton condensates. We have added Figure R3 in Figure 4 and modified related discussion (Pages 10 and 11) to clearly interpret repulsive behavior of polaritons in the revised Manuscript.

Comment 6: *In SI Figure 7, text says: “The diffraction peaks in the XRD pattern of the PDI-O microbelts well match those of the simulated results from the single-crystal diffraction of PDI-O”, but the two XRD spectra don’t look alike at all. What is been compared here?*

Response 6: We thank the reviewer for the comment, which would be helpful for us to better interpret the crystallinity of the PDI-O microbelts. In Supplementary Figure 7, the simulated XRD pattern corresponds to powder samples, and therefore it exhibits the diffraction peaks of all crystal planes that might be detected. For the experimentally measured XRD pattern, however, the obtained diffractions peaks are attributed to the crystal planes parallel to the substrate. As shown in Supplementary Figure 4, the self-assembled PDI-O microbelts are all lying on the substrate due to the small height, and therefore the measured XRD pattern of the as-prepared samples exhibits diffraction peaks corresponding to only one crystal plane, which was ascribed to be (100) crystal plane according to the simulated patterns.

To clearly interpret the XRD data, we have added “*The XRD pattern of the as-prepared microbelts which are all lying on the substrate (Supplementary Fig. 4) exhibits only diffraction peaks corresponding to the (100) crystal plane, which indicate that the direction of the belt thickness is perpendicular to the (100) crystal plane.*” in the discussion of Supplementary Figure 7 in the revised Supplementary Information.

Comment 7: *Have authors tried approaching the threshold for regular photonic*

lasing? It would be interesting to see if they can observe both thresholds or if the material degrades first.

Response 7: Thanks for the comment. We agree with the reviewer that it would be very significant for further understanding polariton lasing if we can observe the second photon lasing threshold. According to the reviewer's suggestion, we have measured input-output curves of the microbelts at high pump power, but unfortunately the PL intensity decrease with increasing pump power due to the material degradation, which inhibits the observation of a second threshold for conventional photon lasing. The threshold for photon lasing can be estimated by measuring the threshold for ASE in the PDI-O microcrystals without cavity effects. Figure R4a is the input-output curves of a PDI-O microcrystal with irregular shape, from which we obtained the threshold for ASE of $\sim 15 \mu\text{J cm}^{-2}$. The input-output curves of the PDI-O microbelt in Figure R4b shows that at such pump power no another threshold appears, and the PL intensity starts to decrease with increasing pump power due to material degradation. We believe that the absence of the second threshold is ascribed to the efficient exciton-polariton strong coupling which transforms excitons into stable polaritons. We have added Figure R4 and related discussion in the revised Supplementary Information as Supplementary Figure 14 to inform readers of this result.

Figure R4 PL intensity as a function of pump fluence in a PDI-O microcrystal with irregular shape (a) and a regular PDI-O microbelt (b). Error bars indicate 95% confidence intervals from five representative measurements.

Comment 8: *Why does the electric profile of Suppl Fig 10 (left image) look the way it*

does, i.e. why strong signal is only observed near the edges? Reconcile them with points 1 and 2.

Response 8: Thanks for this comment which helps us to better present the simulation results. The electric profile was obtained from finite-difference time-domain (FDTD) simulation (FDTD solutions, Lumerical solutions, Inc.). As the system is excited with a dipole source, we used time apodization in the Monitor to filter away short lived transients and get the correct mode profile for the cavity (for more detail, see the description for apodization option of Frequency monitors in the official website of Lumerical: <https://support.lumerical.com>). Because the apodization time width we set was very small (1 fs), the results in Supplementary Fig. 10 represent the instant electric field distributions which evolve with time as light oscillate and leak out of the cavity. Figure R5 shows the electric profile at three different time for a same dipole orientation, from which we can observe an evolution of the electric field distributions and intensities of the cavity mode. To get a more complete mode profile, we have optimized the time apodization (specifically, the apodization center and width) and obtained the results shown in Figure R6, which clearly indicates the cavity effect in the microbelt width direction. This is consistent to the conclusion in responses to point 1 and 2 that the microbelts can form F-P cavities in the direction of the belt width.

In order to avoid possible misleading, we have replaced Supplementary Figure 10 with Figure R6 in the revised Supplementary Information.

Figure R5 Instant electric profile at three different time for a same model geometry. It should be noted that the times were selected so that the three results are electric fields of one and the same cavity mode.

Figure R6 Simulated electric field intensity distribution in an F-P microcavity (in the x-direction) with a dipole source μ (red double-headed arrow) located at its center in different orientation. The dipole is in the x-y plane and the angle between dipole and cavity direction are 90° , 45° , and 0° , respectively.

Comment 9: *How do these results compare to recent cavity free polariton work on crystals from uniaxially aligned oscillators by Meskers, topology work from Yuen-Zhou, single crystal microcavity work including J-aggregates from Bittner, Silva, Spano and others? Authors should consider placing their work in a wider context for readers to gain a deeper appreciation for progress in this field.*

Response 9: Thanks for the suggestion that is very helpful for us to improve the manuscript by giving a more comprehensive introduction for this area.

Meskers's work (eg., *J. Chem. Phys.* **145**, 194703 (2016)) developed a theory for optical properties of polariton involved molecular crystals which is modeled by a cubic lattice of oriented Lorentz oscillators. The authors resolve the issue of the additional boundary conditions by applying a uniform gauge condition, and therefore the experimental reflection spectra can be reproduced accurately by the theory. Meskers have studied the behaviour of incident light in the crystals, which is different from our results about the emission light of molecules within microcavities. Despite of this, Meskers's study is highly instructive of our work as their dipole lattice model is commonly applicable to the molecular crystal systems.

Yuen-Zhou and coworkers have studied plexcitons resulted from the strong coupling between excitons and plasmons in the system containing an organic film patterned on

silver metal. Their theoretical proposal of plexciton topologically protected modes (eg., *Nat. Commun.* **7**, 11783 (2016)) and experimental demonstration of active suprawavelength polaritonic metasurfaces (eg., *Phys. Rev. Lett.* **122**, 173902 (2019)) highlight the potential of organic plexitons as a great platform for the design of exotic modes in strongly coupled light-matter systems. Although the approaches to light confinement are different, Yuen-Zhou's work and our results are both supplements to the study of strong coupling without external cavities.

The theoretical work on single crystal organic microcavities from Bittner, Silva, Spano, La Rocca and others contribute a lot to microscopic understanding of organic microcavities. Our work was partly inspired by some of their results, such as microcavity enhanced exciton coherence length (eg., *J. Chem. Phys.* **142**, 184707 (2015)), vibronically assisted scattering (eg., *Phys. Rev. B* **88**, 075321 (2013); *arXiv:1206.2906v1* (2012)) and adverse effect of orientational disorder on the creation of polariton condensates (eg., *Phys. Chem. Chem. Phys.* **14**, 3226–3233 (2012)). We believe that our work would be a significant validation of their theories.

Following the valuable suggestion of the reviewer, we have revised the manuscript and added a sentence “Recent progresses on cavity free organic polariton systems, such as molecular crystals²⁵ and topological plexitons^{26,27}, offer a strategy to address this issue by realizing BEC in structures with diverse geometries.” in the introduction (Page 4, line 1) to better introduce the research background. Accordingly, we have added related references in the revised Manuscript (*J. Chem. Phys.* **145**, 194703 (2016); *Nat. Commun.* **7**, 11783 (2016); *Phys. Rev. Lett.* **122**, 173902 (2019); *J. Chem. Phys.* **142**, 184707 (2015); *Phys. Chem. Chem. Phys.* **14**, 3226–3233 (2012); *Phys. Rev. B* **88**, 075321 (2013)).

Response to Reviewer 2

General Evaluation: *The authors report on the realization of a polariton condensate at room temperature in so-called microbelt cavities, microcrystals, which form during the self-assembly process and may serve as low- Q microcavities. They find that in these microbelt cavities polariton condensation indeed becomes possible under pulsed excitation. The polariton dipole moment is found to show a strong directionality, which results in a significant dependence of the emission on the polarization and geometric orientation. The authors also find that the momentum of the polaritons depends strongly on the pump power.*

With respect to the validity of the results here, I tend to believe the authors' claim that polariton condensation is actually taking place. The authors show only limited experimental data - including the line width, output intensity and decay time of the mode. However, the momentum redistribution discussed at the end of the manuscript will usually occur only in the condensed regime, which is somewhat reassuring.

Response: We thank the reviewer for the nice summary of our work and the recognition of our experimental findings. We are also very grateful for the insightful comments and constructive suggestions, which have helped us to make a further improvement of our work. In the following, we provide concrete responses to the comments and suggestions point-by-point.

Comment 1: *However, I strongly suggest that the authors either should add more experimental points to the input-output curve shown in figure 3)d) or (if higher cw powers should break the system) instead show an input-output curve for pulsed excitation as well. At current, the input-output curve is not convincing at all.*

Response 1: Thanks for the constructive suggestion. Following the reviewer's comment, we have measured PL intensity and FWHM at a broad range of pump power and plotted the input-output curve as displayed in Figure R7. The PL intensity was plotted in a log-log scale to better display the degree of enhancement. In addition,

we have also added error bars indicating 95% confidence intervals from five representative measurements in Figure R7 to increase the reliability of the experimental results. From Figure R7 we can observe a nonlinear increase of PL intensity and a sharp decrease of FWHM when pump fluence reaches the threshold, which was one of the characteristics of polariton condensation.

Accordingly, we have replaced Figure 3d with Figure R7 in the revised Manuscript.

Figure R7 Emission intensity and FWHM at $k_y = 0$ ($\theta = 0$) as a function of pump fluence. Error bars indicate 95% confidence intervals from five representative measurements.

Comment 2: *I further suggest strongly that the authors show more detailed data on the momentum redistribution for pulsed pumping for different pump powers. If the polariton redistribution in momentum space shows a reasonable dependence on pump power, I would take this as a reasonable experimental signature of condensation taking place.*

Response 2: Thanks for the suggestion, which helps us to better demonstrate the repulsive interactions. Following this suggestion, we have measured angle-resolved PL spectra and the corresponding PL images of a microbelt at different pump powers as shown in Figure R8. With pump power increasing, the wave vector shows a clear increase, accompanied by the increase of the distance between emission position and excitation area, which is reasonable considering the large repulsive interactions at high exciton density. The right panel of Figure R8 plots the horizontal cross-section intensity profiles of the corresponding PL images (black), as well as the wave

function probability density (red) calculated by solving Gross-Pitaevskii equation where a power-dependent Gaussian potential resulting from repulsive interactions is taken into account. The great agreement between the experimental and theoretical results further evidence the polariton-exciton repulsive interactions in the condensates.

Accordingly, we have added Figure R8 in Figure 4 and modified related discussion (Pages 10 and 11) in the revised Manuscript.

Figure R8 Left: angle-resolved PL spectra at different pump powers. Middle: the corresponding PL images of PDI-O microbelts. Scale bar is 10 μm . Right: horizontal cross-section intensity profiles of the corresponding PL images (black) and the wave function probability density calculated by solving Gross-Pitaevskii equation (red).

Comment 3: *In order to establish the material platform in the long run, it will also be necessary to investigate spatial coherence and second-order coherence, but these two points be a topic for future manuscripts.*

Response 3: We appreciate for the reviewer’s helpful comment. The demonstration of the spatial coherence would also provide a more convincing evidence of the formation

of BEC in this work. Therefore, we have carried out a Young's double-slit interferometry experiment (*Phys. Rev. Lett.* **99**, 126403 (2007)) to probe the emergence of spatial coherence. A sketch of the setup is shown in Figure R9a, where the magnification coefficient at real plane 1 is 50, and the slit separation is $60\ \mu\text{m}$. The CCD records the interference pattern of the microbelt's emission passing through the double slit. Figure R9b (R9c) and R9d (R9e) show the recorded interference patterns and the corresponding intensity profile below (above) threshold, respectively. Below threshold, the interference fringes are barely visible. In contrast, above threshold distinct interference fringes are clearly observed with the visibility contrast of 35%, indicating the emergence of spatial coherence of BEC in the microbelt.

Figure R9 **a**, A sketch of the double-slit experiment setup. **b**, Raw image of the interference pattern and **d**, corresponding intensity profile of the microbelt recorded below threshold. **c**, Raw image of the interference pattern and **e**, corresponding intensity profile of the microbelt recorded above threshold. Scale bars are $3\ \mu\text{m}$.

We have added Figure R9b and R9c in Figure 3 and modified the related discussion as

“The condensates feature an emergence of long-range spatial coherence, which was probed by a Young’s double-slit interferometry experiment (Supplementary Fig. 15)⁴⁴. The recorded pattern shows barely visible interference fringes below threshold (Fig. 3f), while above threshold distinct interference fringes are clearly observed with the visibility contrast of 35% (Fig. 3g), indicating the emergence of spatial coherence of BEC in the microribbon.” (Page 9, line 11) in the revised Manuscript. We have also added Figure R9a, Figure R9d and R9e and related discussion as Supplementary Fig. 15 in the revised Supplementary Information. The above-mentioned reference (*Phys. Rev. Lett.* **99**, 126403 (2007)) has also been added in the revised Manuscript.

Comment 4: *With respect to the potential impact of the present manuscript, polariton condensation at low quality factors is certainly of interest to the wider community, especially as it is very difficult to achieve large quality factors for organic materials. In addition to the points just mentioned, although the manuscript certainly provides some interesting aspects, I cannot recommend it for publication in the present form due to some shortcomings that in my opinion put the manuscript below the standards of Nature Communications at current. However, it might be possible to overcome these problems by further changes additions to the manuscript. There are some minor points to be listed later on, but my main reservations with respect to the validity and the impact of the results presented here are twofold. First, for some aspects there is very little quantitative information given. Accordingly, it will be hard for the reader to correctly judge and even reproduce the results presented here. Second, there are some inconsistencies in the explanations given here, which definitely must be taken care of. Let me express my concerns in detail:*

Response 4: We thank the reviewer for the positive evaluation of our work. We are also very grateful for the insightful comments and suggestions, which help us to be more precise about the presentation and interpretation of the scientific rigor and importance of our work.

Comment 4-1: *One of the main selling points of the authors consists of the fact that they are able to realize polariton condensation even in low-quality organic cavities due to the rather large dipole moments they are able to achieve. However, the author never gets to know what a low cavity quality factor actually means in this case. As the cavities used in organic polaritonics usually show low Q factors as compared to epitaxially grown semiconductor microcavities anyway, the reader will need some quantitative information to be able to judge how much of an advancement the results presented here actually are. I am aware that there are plenty of cavity modes supported by the structure used here and measuring the Q factor exactly is a nontrivial task, but this task is indeed easier for low- Q modes. Accordingly the number should be presented. This number will in turn be required, so the reader can judge the importance of the other main result of the authors correctly: Control over the polaritons via the interaction with the exciton reservoir. Indeed, this is an interesting phenomenon, but it is usually considered to be most useful in the case of large- Q cavities, where the distances traveled by the polaritons within their lifetime may become mesoscopic. As the cavity quality factor also determines the polariton lifetime, it is in my opinion necessary to provide the polariton lifetime in the present structure as well, so the reader may judge whether the travel distances that may be achieved are actually long enough for "constructing miniaturized polaritonic devices towards practical applications in photonic circuits" or not.*

Response 4-1: We are grateful for this valuable suggestion. We agree with the reviewer that the Q factor of the microbelt cavity is a very important and necessary information, which not only helps to concretize the significance of our work but also matters for the estimation of polariton lifetime when discussing the polariton travelling distance.

Calculation of Q factor:

Following the reviewer's suggestion, we have calculated the Q factor of the microbelt cavity from the mode spacings in PL spectra. Figure R10 is the PL spectrum of a microbelt excited by 405 nm CW laser, which exhibits multiple resonance peaks. The

Q factor was calculated by $Q = \lambda/\Delta\lambda$, where $\Delta\lambda$ is the full-width at half-maximum (FWHM) of the resonance peaks obtained from the fitted Lorentzian line shapes. We have performed measurements and calculations for dozens of microbelts and found that Q factors of the cavities are in range from 130 to 170 for $\lambda = 585$ nm, which is obviously lower than that of the reported organic microcavities for polariton BEC (eg., $Q \sim 600$ in *Nat. Mater.* **13**, 271-278 (2014)).

Estimation of Polariton lifetimes.

Considering the large photonic fraction at zero wave vector, we used $\tau = Q\lambda/2\pi c$ to calculate polariton lifetime, which was estimated as ~ 50 fs. This result is half smaller than the lifetime of organic polaritons (100 fs) in the work on polariton superfluid (*Nat. Phys.* **13**, 837 (2017)). Nevertheless, compared with the thin-film microcavity, our single crystal microbelt exhibits less structural inhomogeneity, which might lead to a mesoscopic polariton travelling distance within the short polariton lifetime.

According to the reviewer's suggestion, we have added Figure R10 in Supplementary Figure 6 and modified the figure caption and related discussion in the revised Supplementary Information.

Figure R10 PL spectrum of a PDI-O microbelt excited by 405 nm CW laser (black solid line). The dashed red lines represent the fitted Lorentzian line shapes. The overall fitting line is displayed by the red solid line.

We have also added following sentences in the revised Manuscript:

(page 6, line 4) “*The cavity direction is parallel to the ribbon width direction, and the Q factor of the cavity was estimated to be ranged from 130 to 170 (Supplementary Fig.*

6), corresponding to a photon lifetime of ~ 50 fs.”

(page 6, line 19) “which is important for BEC considering the relatively low Q factor of the microribbon cavities.”

(page 10, line 13) “Considering the high structural homogeneity of the single crystal microribbons, polaritons could travel over macroscopic distances within their lifetime which was estimated to be ~ 50 fs.”

Comment 4-2: *Maybe some more critical points are given by inconsistencies within the manuscript. The most important problem given in the manuscript is the fact that the authors never clearly define which cavity mode they are actually interested in. Most of the experimental studies presented here focus on an experimental geometry, where angle-resolved (and therefore momentum-resolved) spectra are shown and the relevant angle is the one between the z-axis and the y-axis, where the angle is chosen such that the zero-angle corresponds to an orientation along the z-axis and the $\sin(\theta)$ values shown e.g. in figures 3 and 4 correspond to the wavevector projection k_y along the y-axis. The authors find that the dispersion shows a minimum for $k_y=0$ and shows a parabolic dispersion for low values of k_y . This is the typical and expected result for a cavity mode, where photons are confined along the z-direction, which will typically be called the cavity direction. Instead, in calculations, e.g. in the supplementary figure 10 in the supplementary material, the authors discuss several different orientations of the dipole within the x-y plane and discuss that this change corresponds to different angles between the dipole and the cavity direction. This means that the cavity direction is now aligned along the x-axis, which would be in line with the depiction given in figure 1)e). This is obviously inconsistent. To complicate matters further, the authors frequently refer to the width and length of the cavity as well as to its short and long axis. The directions of those are never explicitly defined within the manuscript, which will make it impossible for the reader to identify the geometry at hand. The same goes for the comparison between TE and TM modes in the supplementary material, which is a great supporting information, but lacks the information with respect to which axis of the structure TE and TM are defined (or*

equivalently: which cavity mode is considered). A thorough introduction of the exact geometry used will be highly beneficial for the reader.

Response 4-2: Thanks for the comment which reminds us of the possible confusion caused by the description of cavity geometry and the experimental setup. In fact, although the emission was collected from the z-axis, the angle-resolved PL spectra we obtained agreed well with the waveguide cavity model where the cavity direction is parallel to the belt width. This waveguide cavity effect involves the travelling of light in the microbelts by total internal reflection (*Nano. Lett.* **14**, 6564 (2014)). The atomic force microscopy (AFM) measurement results reveal that the thickness of the microbelts is typically ~150 nm (Supplementary Figure 5), which is not sufficient to allow the long range travelling of light in the direction normal to the top facet of the microbelts (z-direction). To examine whether cavity confinement in the z-direction exists in the microbelt, we have measured the reflectivity spectrum of a microbelt at normal incidence. As shown in Figure R11, the reflectivity spectrum exhibits uniform reflectance without any dips, indicating the absence of photon confinement in the z-direction. We attributed this to the relatively low reflectance (25%) for the top and bottom facets of the microbelt, which is possibly due to the leakage of light into the dielectric substrate. To conclude, the optical cavity along the belt width is superior to that in z-direction, and dominates the spectral modulation that we have observed in this work.

Figure R11 Reflectivity spectrum at normal incidence of a PDI-O microbelt.

To provide a thorough introduction of the exact geometry, we have made a clearer illustration of the measurement setup (Figure R12), which shows that in the microbelt the guided light propagates via total internal reflection and reflects back and forth in x-direction. Because the vector of the exciton dipole moment is along the y-direction, the waveguide modes in the microbelts are TE polarized, with the electric field direction parallel to the y-direction. These waveguide cavity modes diffract out of the microbelt at the lateral facets, forming point-like sources that emit photons nondirectionally in the x-z plane (*Nano Lett.* **6**, 2707-2711 (2006)). Part of these cavity photons were collected from the z-direction through an objective. The diffraction only occurs at x-z plane due to the small height of the microbelt, and therefore the wave vector in y-axis, k_y (dependent on angle θ), is not affected, which allows us to obtain E- k_y relationship of the waveguide cavity by directly measuring angle resolved PL at y-z plane.

To provide more informative description of the waveguide-involved cavity effect in the microbelt, we have added the word “waveguide” in front of the terms “cavity” or “F-P cavity” in the revised Manuscript and Supplementary Information. We have also added the information of the thickness of the microbelts and the discussion of cavity direction in the revised Supplementary Information.

Figure R12 Schematic of a PDI-O microbelt along with coordinate axes. The direction of the waveguide F-P cavity is parallel to the x-direction. k_y is the projection of wave vector onto y-axis, which depends upon the emission angle in y-z plane (θ).

We have also replaced Figure 2a with Figure R12 which would be more informative about the measurement geometry and modified the related description (Page 7, line 4)

in the revised Manuscript as “As illustrated in Fig. 2a, the guided light travels along x -direction in the microribbon cavity and diffracts out at the lateral facets, forming point-like sources that emit photons nondirectionally in the x - z plane³⁹. The emitted photons were detected in y - z plane at different angles, which reflect the polariton dispersion because the wave vector component along y -axis (k_y) depends upon the angle of propagation (θ).”. We have also added related reference (*Nano Lett.* **6**, 2707 (2006)) in the revised Manuscript.

Comment 4-3: *A minor point, which is still relevant: The strong directionality of the dipole moment within the microbelt structures effectively renders them microwire-like effective 1d structures. On the page showing figure 12 in the supplementary material, the authors discuss the appearance of several modes with unequal mode spacings. The expected mode spacings for 1d microwires of a given width are well-known. Considering the confusion about the experimental geometry already mentioned, it would be very helpful to show that the bare cavity mode energies, which should be obtainable from the coupled oscillator model already applied by the authors, follow the expected trend for the given microbelt widths. This might be important as some of the cavity modes might not couple efficiently to the excitons due to the strong orientation of their dipole moments. Further, it is known that the polariton spectrum may become quite non-trivial including complicated energy shifts in the presence of relaxation processes (*Phys. Rev. B* **82**, 245315 (2010)), so the authors' claims would become much more substantial if they could provide a clear explanation for the mode spectrum in their structure. Along the same lines, the authors might also want to add an explanation on why polariton condensation does not occur in the polariton mode arising from the lowest energy cavity mode.*

Response 4-3: Thanks for the suggestion. We agree with the reviewer that presenting the calculated bare cavity modes will provide strong evidence supporting the cavity effect in the belt width direction.

1) *Bare cavity modes.*

To calculate the dispersion of the bare waveguide cavity modes, we have fitted the measured dispersion to a coupled harmonic oscillator model which includes the cavity energy, as shown in Figure R13. The microbelts only support modes that simultaneously satisfy the characteristic equation for guided modes and the F-P resonance condition. The following is the analysis of waveguide cavity modes in the microcavity:

For the guided modes which travel inside the microbelt in x-direction by total internal reflection in the zigzag fashion, the sum of all phase shifts after each round trip of the wave must be equal to a multiple of 2π , which can be expressed by the characteristic equation:

$$2k_z d - 2\varphi_t - 2\varphi_b = 2m\pi \quad (1)$$

Where k_z is the wavevector inside the microbelt along the z-axis, and φ_t and φ_b are phase shifts on total reflection from the microbelt's top and bottom facets, respectively. m and d are the mode order of the guided modes and thickness of the microbelt.

The z-component of the mode wave vector k_z can be expressed by:

$$k_z = \sqrt{n_{bg}^2 k_0^2 - \beta^2} \quad (2)$$

Where $k_0 = 2\pi/\lambda$ and n_{bg} are the wave vector in free space and the background refractive index of the microbelt, respectively. $n_{bg}k_0$ and β are the wave vector of the guided modes the propagation constant along the x-axis, respectively. For a microbelt with a width of W , β satisfies:

$$\beta = \frac{N\pi}{W} \quad (3)$$

Where N is integer number for modes.

For TE modes, we can extract from the Fresnel formulas the following expressions for the phase shifts φ_t and φ_b :

$$\tan\varphi_t = \frac{\sqrt{\beta^2 - n_{air}^2 k_0^2}}{\sqrt{n_{bg}^2 k_0^2 - \beta^2}} \quad (4)$$

$$\tan\varphi_b = \frac{\sqrt{\beta^2 - n_{\text{sub}}^2 k_0^2}}{\sqrt{n_{\text{bg}}^2 k_0^2 - \beta^2}} \quad (5)$$

Where n_{air} and n_{sub} are the refractive index of the glass air and substrate, respectively.

For simplification of calculation, we take $n_{\text{air}} = n_{\text{sub}} = 1$.

Because the long length of the microbelt, the y-component of the cavity mode wave vector k_y is free and can be expressed by

$$k_y = k_0 \tan(\arcsin \frac{\sin\theta}{n_{\text{bg}}}) \quad (6)$$

Therefore, the energy of the uncoupled F-P type waveguide cavity mode is

$$E_c = \frac{hc}{2\pi} \sqrt{k_0^2 + k_y^2} \quad (7)$$

Accordingly, we have replaced Supplementary Figure 12 with Figure R13 in the revised Supplementary Information.

2) *the relaxation processes in Phys. Rev. B 82, 245315 (2010).*

The above discussion indicates that the discrete energy modes in the microbelts are resulted from the confinement in the microbelt width direction ($\beta = N\pi/W$), which is rather different from the intracondensate relaxation induced mode separation reported by Wouters (*Phys. Rev. B 82, 245315 (2010)*). We look forward to further experimental study of the intracondensate relaxation process in organic materials by guidance of Wouter's theory in future.

3) *Why polariton condensation does not occur in the polariton mode arising from the lowest energy cavity mode.*

In organic microcavities, polariton ground state can be directly populated from the reservoir with the assistance of molecular vibrations (*Nat. Photonics 13, 378-383 (2019)*). Therefore, polariton condensation usually occurs at polariton modes whose ground states correspond to the energetic separation between the exciton reservoir and vibronic sublevel of the molecular ground state (*Nat. Photonics 4, 371-375 (2010)*). In our microbelts, the strongest vibronic resonance is $S_{10} \rightarrow S_{01}$ transition and therefore polariton condensation only occurs near the energy of $S_{10} \rightarrow S_{01}$ transition of PDI-O molecules.

According to the suggestions of the reviewer, we have added sentences “*The polariton emission is located near 580 nm, which corresponds to $S_{10} \rightarrow S_{01}$ transition of PDI-O molecules, indicating molecular vibration-assisted population of polaritons from the exciton reservoir.*” (page 7 line 21) and “*Condensation preferentially occurs at this mode due to efficient molecular vibration-assisted population process.*” (page 8 line 22) in the revised Manuscript.

Figure R13 AR- μ -PL spectra of microbelts with different width (W). The dispersions of polariton modes (red dashed lines) and bare cavity modes (black dashed lines) are obtained from fitting the measured dispersion to a coupled harmonic oscillator model. Red solid lines represent exciton energy.

Comment 5: *With respect to the state of the art in the field, I am aware that the field is large and it is impossible to cite every single paper that is relevant with respect to the topic at hand, but in my opinion there are some shortcomings with respect to adequately referencing the work of others working in the field:*

The authors utilize the idea that repulsive interactions between polaritons and

excitons and polaritons result in a controllable polariton emission pattern in momentum space. The explanation for this effect was presented first in an important paper by Wouters et al. (Phys. Rev. B 77, 115340 (2008)), which definitely deserves to be cited in the present manuscript. Further, as mentioned above, Phys. Rev. B 82, 245315 (2010) might be a relevant paper here.

Response 5: Thanks for the suggestion. Accordingly, we have double checked and revised the references. The references suggested by the reviewer are quite relevant to our work, especially the work on the theory of the spatial profile and the spectral properties of polariton condensates, which have helped us to interpret the momentum redistribution phenomenon with dissipative Gross-Pitaevskii (GP) equation. We have added the references suggested by the reviewer in the revised Manuscript. (*Phys. Rev. B 77, 115340 (2008); Phys. Rev. B 82, 245315 (2010)*).

Comment 6: *Some additional minor points with respect to clarity and presentation:*

- The dashed lines that are supposed to show the fitted polariton dispersions in figures 3)a) to c) are almost impossible to see. The authors might want to use thicker lines and/or a different color.

- Within the main text, the authors highlight that the main difference between the spectra shown in figure 2 and 3 is the usage of cw versus pulsed excitation. As this is quite important information, it might improve the ease of reading the manuscript if this information was added to the figure captions as well.

- On page 3, the authors judge that "planar microcavities tend to have large footprints". That statement seems too vague to me to make sense for a general readership.

- The authors frequently show angle-resolved emission spectra. It would be helpful for the reader to directly convert the value shown to the wavevector along the y-direction.

Response 6: We thank the reviewer for pointing out these issues and giving helpful suggestions. We have revised the manuscript according to the reviewer's suggestion.

(1) -We have replaced the fitted polariton dispersions curves in Figure 3a-c with

thicker dashed lines for clarity.

(2) -We have improved our experimental setup so that we can detect much weaker signals from the samples. We have replaced Figure 3a-c with Figure R14, which exhibits more clear dispersion modes at pump power below the threshold. We have added a sentence “*The microribbon was excited by a 405 nm CW laser.*” in the caption of Figure 2 and a sentence “*The microribbon was pumped by a pulsed laser (400 nm, 150 fs, 1 kHz).*” in the caption of Figure 3 in the revised Manuscript. We have also modified the related discussion as “*At low pump fluence ($1.8 \mu\text{J cm}^{-2}$), PL spectrum shows emission of multiple polariton modes (Fig.3a), indicating that these polariton branches are all populated.*” (page 8, line 16). Accordingly, we have replaced Figure 2b with the spectra corresponding to the PDI-O microbelt investigated in Figure 3a.

Figure R14 Angle-resolved spectra of the PDI-O microbelt measured at $0.8 P_{\text{th}}$, $1.0 P_{\text{th}}$, and $1.5 P_{\text{th}}$, respectively.

(3) -We agree with the reviewer that the original statement dose not clearly represent our argument that the planar microcavities would occupy large area on the chip, which limits the integration density. Accordingly, we have modified the statement as “*Such planar (two-dimensional) structures tend to have large lateral device footprints and complication to guide EP fluids, which restricts the development of organic polaritonic devices with compactness and integrability.*” (Page 3, line 20) in the revised manuscript.

(4) -We have converted emission angles to wave vectors in the revised manuscript using $k_y = k \tan(\arcsin(\sin\theta/n_{\text{bg}}))$.

Comment 7: *In summary, I cannot recommend the current manuscript for publication in Nature Communications, but a significant revision might be able to meet the journal standards.*

Response 7: We are very grateful for the insightful comments and constructive suggestions. We believe that we have addressed some serious problems in the manuscript and we wish that our alterations and response make sense to you the reviewer.

Response to Reviewer 3

General Evaluation: *The manuscript by Zhao and co-workers claims observation of room temperature polariton Bose-Einstein condensation in organic single-crystal “microbelt” cavities. Such “microbelts” are formed by the organic compound PDI-O through a self-assembly process, and function as Fabry-Pérot microcavities. The PDI-O molecules are densely packed and arranged in such a way in the microcavities that the structure is capable of supporting large densities of Frenkel excitons with large transition dipole moments. Evidence of strong coupling is obtained through angle-resolved photoluminescence spectroscopy, which allows the authors to extract dispersion curves for the FP modes, and fit with a simple model. Evidence of Bose-Einstein condensation is the observation of spectral narrowing and lifetime decrease for one of the FP cavity modes, as a function of pump fluence. The authors also claim the ability to manipulate the condensate, evidenced in the evolution of the emission spatial profile with pump fluence, which suggests repulsion of excitons in the illumination area.*

I find that more clarification is needed for the evidence presented, before I can recommend publication. The following comments must be addressed:

Response: We thank the reviewer for the nice summary of our work. We are also very grateful for the insightful comments and constructive suggestions, which have helped us to make a further improvement of our work. In the following, we provide concrete responses to the comments and suggestions point-by-point.

Comment 1: *The term microbelt is somewhat misleading, as it evokes a circular closed structure. I suggest calling it something else, like ‘nanowaveguide’ or something else that reflects the 1D linear geometry.*

Response 1: Thanks for the helpful suggestion. According to the reviewer's suggestion, we have replaced “microbelt” with “microribbon” in the revised manuscript and Supplementary Information to avoid possible misunderstanding.

Comment 2: *In the angle-resolved PL maps, I would expect to see discrete islands in the energy X momentum space, corresponding to the different longitudinal FP modes. I.e., for each N , which from eq. (4) may corresponds to the transverse mode order, rather than a continuous branch as a function of θ , I would expect to see islands. Why is it not the case, is it an artifact of the plot?*

Response 2: Thanks for the comment which reminds us of confusing description of cavity geometry and the experimental setup. As illustrated in Figure R15, in the microribbon the guided light propagates via total internal reflection and reflects back and forth in the x -direction. These waveguide cavity modes diffract out of the microribbon at the lateral facets, forming point-like sources that emit photons nondirectionally in the x - z plane (*Nano Lett.* **6**, 2707-2711 (2006)). The diffraction only occurs at x - z plane due to the small thickness of the microribbon, and therefore the wave vector in y -axis k_y (dependent on angle θ) is not affected, which allows us to obtain E - k_y relationship of the waveguide cavity by directly measuring angle resolved PL at y - z plane. The cavity modes component wave vectors k_x and k_z can only be discrete values because they are confined by the finite width and thickness of the microribbon, respectively. However, k_y is free because the microribbon can be regarded as having an infinite dimension in y -axis, which means that for each N , the values of k_y are not limited. Specifically, for each N , there is only one set of values for k_x and k_z , and the relationship between k_y and $\sin\theta$ is $k_y = k_{xy}\tan(\arcsin(\sin\theta/n_{bg}))/n_{bg}$, where k_{xy} is the vector sum of k_x and k_z . Therefore, the dispersion (E - k_y) obtained from the angle-resolved PL maps (λ - $\sin\theta$) are continuous curves. The angle-resolved PL maps are measured by the experimental setup illustrated in Supplementary Figure 11, and we have not done any modifications of the experimental data.

We have also replaced Figure 2a with Figure R15 which would be more informative about the measurement geometry and modified the related description (Page 7, line 4) in the revised Manuscript as “As illustrated in Fig. 2a, the guided light travels along x -direction in the microribbon cavity and diffracts out at the lateral facets, forming

point-like sources that emit photons nondirectionally in the x - z plane³⁹. The emitted photons were detected in y - z plane at different angles, which reflect the polariton dispersion because the wave vector component along y -axis (k_y) depends upon the angle of propagation (θ).”. We have also added related reference (*Nano Lett.* **6**, 2707 (2006)) in the revised Manuscript.

Figure R15 Schematic of the measurement of dispersion of a PDI-O microribbon waveguide cavity. The direction of the waveguide F-P cavity is parallel to the x -direction. k_y is the projection of wave vector onto y -axis, which depends upon the emission angle in y - z plane (θ).

Comment 3: *It is not clear where the supplementary expressions (2) & (3) for the FP cavity mode dispersion came from. This needs to be clarified, since a fit to the angle-resolved PL with this model is used to justify strong coupling. It would probably be very helpful also if the authors could present the dispersion curves for the empty cavity, to make the “repulsion-like” behavior, which can be a signature of strong coupling, more obvious.*

Response 3: Thanks for the comment. Equation (2) in the original Supplementary Information is derived from the characteristic equation of guided modes in symmetric slab dielectric waveguides (eg. Kogelnik H. (1975) *Theory of Dielectric Waveguides*. In: Tamir T. (eds) *Integrated Optics. Topics in Applied Physics*, vol 7. Springer, Berlin, Heidelberg.). Here the guided modes are propagating in the x -direction (the microribbon width direction). Equation (3) in the original Supplementary Information is the relationship between the energy and wave vector of the uncoupled F-P type

waveguide cavity mode. To obtain the dispersion curves of the empty cavity, we have performed a fitting to the measured polariton dispersion using the coupled oscillator model. The calculated polariton dispersion curves agree with the experimental angle-resolved PL spectra (Figure R16), indicating the existence of polariton modes resulted from strong coupling between excitons and waveguide cavity modes in the microribbon.

According to the reviewer's suggestion, we have modified related discussion in the revised Supplementary Information as following:

For the guided modes which travel inside the microribbon in x-direction by total internal reflection in the zigzag fashion, the sum of all phase shifts after each round trip of the wave must be equal to a multiple of 2π , which can be expressed by the characteristic equation⁷:

$$2k_z d - 2\varphi_t - 2\varphi_b = 2m\pi \quad (8)$$

Where k_z is the wavevector inside the microribbon along the z-axis, and φ_t and φ_b are phase shifts on total reflection from the microribbon's top and bottom facets, respectively. m and d are the mode order of the guided modes and thickness of the microribbon.

The z-component of the mode wave vector k_z can be expressed by:

$$k_z = \sqrt{n_{bg}^2 k_0^2 - \beta^2} \quad (9)$$

Where $k_0 = 2\pi/\lambda$ and n_{bg} are the wave vector in free space and the background refractive index of the microribbon, respectively. $n_{bg}k_0$ and β are the wave vector of the guided modes the propagation constant along the x-axis, respectively. For a microribbon with a width of W , β satisfies:

$$\beta = \frac{N\pi}{W} \quad (10)$$

Where N is integer number for modes.

For TE modes, we can extract from the Fresnel formulas the following expressions for the phase shifts φ_t and φ_b :

$$\tan\varphi_t = \frac{\sqrt{\beta^2 - n_{\text{air}}^2 k_0^2}}{\sqrt{n_{\text{bg}}^2 k_0^2 - \beta^2}} \quad (11)$$

$$\tan\varphi_b = \frac{\sqrt{\beta^2 - n_{\text{sub}}^2 k_0^2}}{\sqrt{n_{\text{bg}}^2 k_0^2 - \beta^2}} \quad (12)$$

Where n_{air} and n_{sub} are the refractive index of the glass air and substrate, respectively.

For simplification of calculation, we take $n_{\text{air}} = n_{\text{sub}} = 1$.

Because the long length of the microribbon, the y-component of the cavity mode wave vector k_y is free and can be expressed by⁸

$$k_y = k_0 \tan(\arcsin \frac{\sin\theta}{n_{\text{bg}}}) \quad (13)$$

Therefore, the energy of the uncoupled F-P type waveguide cavity mode is

$$E_c = \frac{hc}{2\pi} \sqrt{k_0^2 + k_y^2} \quad (14)$$

Figure R16 AR- μ -PL spectra of microribbons with different width (W). The dispersions of polariton modes (red dashed lines) and bare cavity modes (black dashed lines) are obtained from fitting the measured dispersion to a coupled harmonic oscillator model. Red solid lines represent exciton energy.

Accordingly, the dispersion curves of the uncoupled cavity have also been added in Supplementary Figure 12, as shown in Figure R16.

We have also added above-mentioned references in the revised Supplementary Information (Kogelnik H. Theory of Dielectric Waveguides. In: Tamir T. (eds) Integrated Optics. Topics in Applied Physics (Springer, Berlin, 1975); Lidzey D.G., Coles D.M. Strong Coupling in Organic and Hybrid-Semiconductor Microcavity Structures. In: Organic and Hybrid Photonic Crystals. (Springer, Cham, 2015)).

Comment 4: *The intensity of the PL peaks decreases considerably at higher energies, where the large refractive index increase is observed. How well are the FP peak positions known, in this case? For instance, if a fit was used, error bars indicating the 95 % confidence intervals for the expected peak positions should be included in the graph.*

Figure R17 a, PL spectrum of the PDI-O microribbon. The dashed lines are fitted Lorentzian line shapes to measure the mode spacing $\Delta\lambda$ for calculation of refractive index. Error bars indicate the 95% confidence intervals of the fitted peak positions. **b**, Refractive index derived from the coupled oscillator model (red line) and calculated from experimental data (black dot). Error bars indicate the 95% confidence intervals from 5 representative measurements.

Response 4: Thanks for the suggestion, which helps us to increase the rigorousness and validity of our experimental results. The peak positions in PL spectra were obtained by fitting the experimental results to the multiple Lorentzian line shapes. According to the reviewer's suggestion, we have added error bars indicating the 95%

confidence intervals of the fitted peak positions in PL spectra, as shown in Figure R17(a). To increase the reliability of the experimental results, we have performed five measurements of PL spectra for the calculation of refractive index, and added error bars indicating the 95% confidence intervals from 5 representative measurements in the plot (Figure R17(b)).

Accordingly, we have replaced Figure 2(c) and 2(d) with Figure R17(a) and R17(b), respectively, in the revised Manuscript.

Comment 5: *Eq. (4) in the supplement is a measure of the waveguide group index, which is not a function of the material and the waveguide. Waveguide dispersion alone can lead to large group indices. Can the authors calculate the expected group index due to just waveguiding, considering an FP resonator (or waveguide) with the background index, or a proper refractive index extracted at long wavelengths (i.e., far from the exciton)?*

Response 5: Thanks for the comment. The facile method suggested by the reviewer for measuring the background index of the waveguide cavity is applicable for our PDI-O microribbons thanks to their broad PL spectra. Accordingly, we have measured the angle-resolved PL spectra of the PDI-O microribbon in the wavelength range of 620–680 nm, as shown in Figure R18. Because the photon energies are far away from the exciton energy, the measured dispersion corresponds to the uncoupled waveguide cavity modes. Using the equations discussed in **Response 3**, we can give a best fit to the measured results with fitting background index $n_{bg} = 2.2$.

We have added Figure R18 and related discussion in the revised Supplementary Information as Supplementary Figure 13. Accordingly, we have modified the discussion (page 8, line 5) as “*By measuring the mode spacing and using the coupled oscillator model with fixed fitting parameters, we calculated the refractive index (n) of the microribbon (see Supplementary Section 2) and plotted n as a function of wavelength in Fig. 2d. The calculated refractive index is approximately equal to the background refractive index (Supplementary Fig.) at longer wavelength and shows a*

remarkable increase at wavelengths close to the excitonic resonance of PDI-O (525 nm).” in the revised Manuscript.

Figure R18, AR μ -PL spectrum of the PDI-O microribbon with width of 4.8 μm . Dashed curves are fitted dispersions of uncoupled waveguide cavity modes.

Comment 6: *The linewidth narrowing observed in Fig. 3d is small. Can the authors report the uncertainties associated with the reported linewidths? I also suggest plotting the PL intensity in log scale, so readers can appreciate the degree of enhancement.*

Response 6: Thanks for the suggestion, which helps us to increase the confidence and validity of our experimental results and present these results in a better form for analysis. Following the reviewer’s suggestion, we have performed five measurements of PL intensity and linewidth for each pump fluence to obtain the uncertainties of the experimental results. The measured PL intensity and FWHM versus pump fluence are plotted in Figure R19 which includes error bars indicating the 95% confidence intervals from 5 representative measurements. The PL intensity was plotted in a log-log scale for a better representation of the degree of enhancement. Figure R19 shows that when pump fluence reaches threshold, PL intensity increases 2-3 order of magnitude, and FWHM collapse from 1.6 nm to 0.8 nm, which indicates the onset of polariton condensates.

According to the reviewer’s suggestion, we have replaced Figure 3d with Figure R19 in the revised Manuscript.

Figure R19 Emission intensity and FWHM at $k_y = 0$ ($\theta = 0$) as a function of pump fluence. Error bars indicate 95% confidence intervals from five representative measurements.

Comment 7: *Still regarding the results in Fig. 3, more convincing evidence of BEC formation would be through interferometric measurements of the spatial and temporal coherence. Have the authors tried to perform such measurements.*

Figure R20 **a**, A sketch of the double-slit experiment setup. **b**, Raw image of the interference pattern and **d**, corresponding intensity profile of the microribbon recorded below threshold. **c**, Raw image of the interference pattern and **e**,

corresponding intensity profile of the microribbon recorded above threshold. Scale bars are 3 μm .

Response 7: We appreciate for the reviewer's insightful comment which helps us to improve our manuscript by demonstrating the long-range coherence as it is an important characteristic of BEC.

We have carried out a Young's double-slit interferometry experiment (*Phys. Rev. Lett.* **99**, 126403 (2007)) to probe the emergence of spatial coherence in the microribbon. A sketch of the setup is shown in Figure R20a, where the magnification coefficient at real plane 1 is 50, and the slit separation is 60 μm . The CCD records the interference pattern of the microribbon's emission passing through the double slit. Figure R20b (R20c) and R20d (R20e) show the recorded interference patterns and the corresponding intensity profile below (above) threshold, respectively. Below threshold, the interference fringes are barely visible. In contrast, above threshold distinct interference fringes are clearly observed with the visibility contrast of 35%, indicating the emergence spatial coherence of BEC in the microribbon.

Apart from Young's double-slit interferometry experiment, coherence can also be probed using a Michelson interferometer with one of the arms replaced by a retroreflector. Moreover, the Michelson interferometry experiments provide an approach to investigation temporal coherence by changing the delay time between the two arms of the interferometer (*ACS Photonics* **7**, 384 (2019)). We have tried to measure the coherence time of BEC in the microribbon with a Michelson interferometry experiment setup. Unfortunately, we were unable to observe interference fringes in the combined images duo to the very small (one-dimensional) polariton emission overlap area of the two centro-symmetrically inverted images of the microribbon. Figure R21a shows the image of a microribbon together with its inverted image, from which we can find that only one edge of the images overlaps with each other. This one-dimensional pattern makes the interference fringes not readily observable. To examine the usability of the home-built Michelson interferometry experiment setup, we have recorded the images of the same

microribbon without filtering the excitation laser, as shown in Figure R21b. The obvious interference fringes of the excitation laser (150 fs, 1k Hz) indicates that interference condition is satisfied with our setup. Therefore, we attribute the failure of the observation of interference fringes to the one-dimensional nature of polariton condensates in the microribbons. Although we were unable to investigate the temporal coherence in the microribbons, we have provided an evidence of spatial coherence from Young's double-slit interferometry experiment results.

Figure R21 **a**, The image of a microribbon together with its inverted image. **b**, The images of the same microribbon without filtering the excitation laser. Scale bars are 5 μm .

We have added Figure R20b and R20c in Figure 3 and related discussion “*The condensates feature an emergence of long-range spatial coherence, which was probed by a Young’s double-slit interferometry experiment (Supplementary Fig. 15)⁴⁴. The recorded pattern shows barely visible interference fringes below threshold (Fig. 4f), while above threshold distinct interference fringes are clearly observed with the visibility contrast of 35% (Fig. 4g), indicating the emergence of spatial coherence of BEC in the microribbon.*” (page 9, line 11) in the revised Manuscript to demonstrate the spatial coherence of BEC in the microribbons. We also added Figure R20a, Figure R20d and R20e and related discussion in the revised Supplementary Information. The related reference (*Phys. Rev. Lett.* **99**, 126403 (2007)) was also added in the revised Manuscript.

Comment 8: *Regarding results in Fig. 3e: reduction of lifetime can be observed in ASE in laser cavities with emitter ensembles, in going from uncorrelated, individual*

emission to collective emission. Can the authors provide any evidence the observed lifetime reduction is due to stimulated scattering, and not due to the mentioned mechanism?

Response 8: Thanks for the comment, which would be helpful for us to better demonstrate the stimulated scattering mechanism in the microribbons. In the work on BEC in microcavities composed of a single film of TDAF sandwiched between two DBRs (*Nat. Mater.* **13**, 271-278 (2014)), authors have compared PL lifetimes of microcavities and bare TDAF thin films. When BEC occurs in microcavities, PL lifetime collapses to <30 ps, while for bare thin films, PL lifetime in the presence of stimulated emission was 133 ps. This result indicates that stimulated scattering process depletes reservoir population more rapidly than stimulated emission. It has been reported that in PDI-doped polymer waveguide (*ACS Photonics* **4**, 114-120 (2017)), PL lifetime at pump power above ASE threshold is 120 ps, which is larger than 47 ps of our microribbon. Considering the non-radiative decay resulted from H-aggregation in their waveguide structure, we suppose that stimulated emission involved PL lifetime in our microribbons would be even longer than 120 ps. Therefore, the very short PL lifetime of 47 ps is attributed to an ultrafast stimulated scattering process.

We have modified related discussion as “*The scattering mechanism is verified by time-resolved PL measurement results in Fig. 3e, which show that the emission lifetime decreases from 277 ps to 47 ps when pump fluence is increased above the threshold. Such lifetime is much shorter than that of PDI molecules undergoing stimulated emission, indicating a transition from exciton reservoir dynamics to an ultrafast decay process corresponding to the stimulated scattering from the exciton reservoir to the condensate*” (page 9, line 5) and added the above-mentioned reference in the revised Manuscript (*ACS Photonics* **4**, 114-120 (2017)).

Comment 9: *The explanation for Fig.4, regarding the excitonic repulsion, is very qualitative and should be better supported. Have the authors attempted to simulate*

the behavior of Fig. 4 with the Gross-Pitaevskii model shown in the supplement? Doing this could provide better evidence of the achievement of BEC and of the exciton repulsion as claimed.

Response 9: We thank the reviewer for the instructive suggestion. The quantitative interpretation of the experimental results would better verify the existence of the repulsion interactions, which in turn helps to evidence the realization of polariton BEC.

Figure R22 Left: the corresponding PL images of PDI-O microribbons. Scale bar is $10 \mu\text{m}$. Right: horizontal cross-section intensity profiles of the PL images (black) and the wave function probability density calculated by solving Gross-Pitaevskii equation (red).

Due to the small width and thickness of the microribbon, polaritons are confined in two dimensions, and therefore we can use a one-dimensional (1D) model to describe the polariton condensate with interaction potential. Accordingly, we have collected the PL images of a PDI-O microribbon at different pump powers (Left panel in Figure R22), from which we have extracted the horizontal cross-section PL intensity profiles (black solid lines in right panel). These experimental data were fitted according to a

1D Gross-Pitaevskii equation:

$$\left\{ -\frac{\hbar^2}{2m} \nabla^2 + V_{\text{res}}(r) + g|\psi(r)|^2 \right\} \psi(r) = \nu\psi(r) \quad (15)$$

Where the repulsion between polaritons and excitons $V_{\text{res}}(r)$ has a Gaussian profile and pump power-dependent intensity. The red solid lines in Figure R22 are calculated wave function probabilities of polaritons, which provide a good fit to the experimental data, confirming the existence of the repulsive interactions in the microribbon. The fitted polariton effective mass (m : $\sim 2 \times 10^{-35}$ kg) and repulsive interaction potential ($V_{\text{res}}(r)$: in order of 0.001 eV) are consistent to the previous report on organic polariton condensates (*Nat. Mater.* **13**, 271-278 (2014)).

According to the reviewer's suggestion, we have added Figure R22 in Figure 4 and modified the related discussion in the revised Manuscript (Page 11). We have also modified the discussion on repulsive interactions (Supplementary Section 3) in the revised Supplementary Information.

REVIEWER COMMENTS

Reviewer #1 (Remarks to the Author):

I appreciate the effort put in by authors to respond to the list of questions by all three reviewers. Authors's have responded to questions satisfactorily and the revised manuscript is much improved. I recommend its publication in Nat Comm with few minor corrections that I'd like authors to consider - see below.

1. There are two X-Y axis shown in figure 10 (SI). Consider consolidating it into one and represent dipole direction better for clarity.
2. I like the PL measurement setup in the SI and wonder if the figure can be moved to the manuscript. It's not necessary or critical and I'll leave it on authors to think if there is merit in that.
3. I find fig 2a helpful in demonstrating F-P geometry and emission, however, the figure is still a bit dense especially close to zoomed section. I believe authors want to represent conical spread of emission from point dipole and I can figure that reading the manuscript and revision but for first time reader this maybe a bit complicated. Authors should consider making this figure simpler, yet engaging, for the reader.

Reviewer #2 (Remarks to the Author):

The authors present a revised version on their manuscript on polariton condensation in organic microbelt (or in the revised version: microribbon) cavities. In the first round of refereeing three referee reports were returned. The questions raised in all three referee reports were rather similar. So was the general feeling of the referees about whether the manuscript at hand is suitable for Nature Communications: The referees mostly agreed that the topic at hand is suitable for Nature Communications, but the manuscript certainly requires additional clarifications and experimental data.

Accordingly, in the following I will mainly focus on the question whether the authors succeeded in clarifying the points raised by the referees. In my opinion, the following points were the most important ones:

1) The experimental geometry

All referees agreed that the experimental geometry is not described well and may potentially give rise to misunderstandings. Most importantly, a clear description of the propagation direction of polaritons and a clear discussion of which axes are used to record the dispersion (and why) was missing. This point has definitely improved substantially. The explanation that the emission in the x-z-plane is nondirectional due to the small extension of the system along these directions is highly relevant and in my opinion absolutely necessary to understand the manuscript at hand. In my opinion the author response to this point is reasonable and convincing.

2) validity of polariton condensation

All referees implied that the authors might be able to find more convincing indications for polariton condensation actually taking place in the sample. The referees suggested that investigations of the spatial coherence of the emission, a more thorough and quantitative treatment of the spatial redistribution of polaritons due to interactions and a more conclusive input-output curve might be helpful here. In my opinion, the input-output curve and the discussion on the interaction-induced spatial redistribution are convincing now. The additional discussion on spatial coherence is nice,

but the contrast observed is not too great. This is somewhat expected due to the typical inhomogeneity of the system. However, as the discussion about the exact experimental geometry was a major point of confusion in the initial manuscript, I would suggest to add axis descriptions and labels to the axes of figures 3)f) and 3)g), so the reader may immediately find the y-direction in the figure. Besides that, I consider the response of the authors with respect to this point satisfactory.

3) Cavity modes

There have been several comments from the referees requesting that the authors clarify the mode structure of their cavity modes, both with respect to the quality factor and the polarization properties of the individual modes. In my opinion the authors' response to these points is reasonable.

In my opinion, these were the main points raised. A part of the referees had more specific questions, e.g. on the XRD spectra, where the referee who raised this point will be more competent to judge whether the reply is satisfactory than I am. All referees also seemed to agree that the topic investigated here is relevant and interesting enough for the wider audience of Nature Communications. Accordingly, assuming that the results presented here are free from unintentionally or deliberately introduced errors, I recommend the revised manuscript for publication in Nature Communications.

Reviewer #3 (Remarks to the Author):

The authors made a number of changes to the text and provided reasonable justifications to most of my comments. However, I still have a few issues, particularly with the modeling, which prevent me from recommending publication without further revision.

Generally, I feel that modeling the ribbon as an FP cavity along x is confusing, because it is much more natural to think of the nanoribbons as waveguides along the y-direction, supporting multiple transverse modes (i.e., with characteristic field distributions in the xz-plane). There are many ways to model this - numerical methods are generally necessary to be exact, however simplified methods with semi-analytical expressions also exist (see for instance Marcatili's classical paper on rectangular dielectric waveguides, The Bell System Technical Journal Volume: 48, Issue: 7, Sept. 1969) which would be adequate to model the nanoribbons.

I also have a bit of an issue with calling the nanoribbons FP cavities in the x-direction, because they extend very far in the y-direction comparatively. In particular, the FP mode simulation discussed in Supplementary Fig. 10 might not be a good representation of the physical structure, because it appears that only a small section of a nanoribbon was simulated, and there is no information about the boundary conditions used. This could have artificially selected the TE modes that were presented. The simulation here should be re-run with 'open boundary'-type boundary conditions, such as Perfectly Matched Layers, or at least Bloch-type boundary conditions.

There are also other issues in the responses to some of my original comments:

Response to comment 2:

I disagree that k_y can take infinite values, because the ribbon, though long, is not infinite. There should be resonances according to (roughly) $k_y L = 2\pi n$, with k_y for a particular m, and L the length (extent in the y-direction) of the ribbon. In other words, the ribbon forms a multimode Fabry-Perot resonator along the y-direction. If there are no experimentally visible resonances, perhaps the modal reflectivity at the two ends of the nanoribbon is extremely low, or maybe the spectral resolution is not sufficiently high to resolve such resonances. The authors should verify that that is the case, or offer a better explanation for which such resonances are not observed.

Response to comment 3:

Equation (10) in the response is not exact, though it may work as an approximation. In reality, eqs. (11), (12) and (8) have to be solved simultaneously in order to yield a value of β . These expressions, however, are only strictly valid for a slab that extends infinitely in the z direction, which is not the case here. The authors should clarify this in the text.

Also, an important simplification that was done in eqs. (11) and (12) was that $n_{\text{air}} = n_{\text{sub}} = 1$. In reality, the slab has a higher n_{sub} , forming an asymmetric waveguide. This gives a more limited spectrum than the symmetric case. For a self-consistent model, have the authors verified that all the relevant slab modes still exist in the asymmetric structure?

Response to comment 4:

In Fig. R17 b, the authors plotted error bars indicating 95 % confidence intervals from 5 representative measurements. The error bars to be plotted should be for the refractive index uncertainty, propagated from the 95 % confidence intervals from the fit in Fig. R17a.

Response to comment 6:

As in the response to comment 5, the authors should plot the 95 % confidence intervals for the FWHM obtained from the fits to the data.

Lastly, regarding the vibron-assisted scattering being the reason for the polariton condensation at the 580 nm wavelength, my understanding is that the polariton should be energetically separated from the exciton reservoir by the vibron energy, which is probably of the order of hundreds of meV. I believe the claim in the main text that "The polariton emission is located near 580 nm, which corresponds to S10→S01 transition of PDI-O", is not exactly the same. Have the authors characterized the vibronic energies of the PDI-O molecules? I think this information should be provided in the text and better explained, in order to justify the vibron-assisted scattering claim.

Response to Reviewer 1

General Evaluation: *I appreciate the effort put in by authors to respond to the list of questions by all three reviewers. Authors's have responded to questions satisfactorily and the revised manuscript is much improved. I recommend its publication in Nat Comm with few minor corrections that I'd like authors to consider - see below.*

Response: We thank the reviewer for the recognition of our revised Manuscript and recommendation of its publication. In the following, we provide concrete responses to the reviewer's suggestions point-by-point.

Comment 1: *There are two X-Y axis shown in figure 10 (SI). Consider consolidating it into one and represent dipole direction better for clarity.*

Response 1: We thank the reviewer for the suggestion that would help us to better represent the simulation results. According to this suggestion and also the comments of the Reviewer 3, we have modified the simulation study. First, we have simulated the emission profile of a dipole (Figure R1a), which is torus-shaped and the emission intensity is strongest in the direction perpendicular to the dipole axis, indicating that the coupling between the dipole and cavity modes is orientation dependent. Then we have simulated the electric field distribution in a PDI-O single-crystal microribbon, which was modeled as a $15\ \mu\text{m} \times 2\ \mu\text{m} \times 200\ \text{nm}$ cuboid with a dipole source located at its center (Figure R1b). The simulated results show that for a microribbon with the dipole source parallel to the y-direction, the cavity modes in the x-direction are preferentially formed due to the directionality of the dipole emission. This suggests a large coupling strength between the dipole and cavity modes in the microribbon, and therefore in our single-crystal PDI-O microribbons all the Frenkel excitons can efficiently undergo strong coupling with cavity photons, which facilitates the realization of polariton condensation.

Figure R1 a, Stimulated emission profile of a dipole in the three-dimensional space (left) and xz or yz-plane (right). **b**, Simulated electric field intensity distribution in a microribbon with a dipole source μ (red double-headed arrow) located at its center. The dipole is parallel to y-direction.

To better represent our simulation results, we have replaced Supplementary 10 with Figure R1 in the revised Supplementary Information. Accordingly, we have modified the related discussion as “*This is further supported by the finite-difference time-domain (FDTD) simulation results (Supplementary Fig. 10), which show that for a microribbon with a dipole source perpendicular to its long axis, the cavity modes in the width direction are preferentially formed due to the directionality of the dipole emission.*” and “*The emission pattern of a dipole is torus-shaped and the emission intensity is strongest in the direction perpendicular to the dipole axis, which indicates that the coupling between the dipole and cavity modes is orientation dependent. Supplementary Fig. 10b shows that when the dipole source is along the y-direction,*

the microribbon supports the cavity modes in the x -direction due to the directionality of the dipole emission, which suggests a large coupling strength between the dipole and cavity modes in the microribbon. Therefore, in our single-crystal PDI-O microribbons, all the Frenkel excitons can efficiently undergo strong coupling with cavity photons, which facilitates the realization of polariton condensation.” in the revised Supplementary Information.

Comment 2: I like the PL measurement setup in the SI and wonder if the figure can be moved to the manuscript. It's not necessary or critical and I'll leave it on authors to think if there is merit in that.

Response 2: Thanks for the reviewer’s suggestion. We agree that the schematic of the angle-resolved PL measurement setup in Supplementary Figure 13 would help to better represent the experimental geometry and principle. Following the reviewer’s suggestion, we have moved the figure to Figure 2 in the revised Manuscript, as shown in Figure R2. Accordingly, we have moved the description of the measurement setup to Method in the revised Supplementary Information.

Figure R2 a, Schematic of the experimental setup for angle-resolved

microphotoluminescence (AR μ -PL) measurement. **b**, Schematic of a PDI-O microribbon along with coordinate axes. **c**, AR μ -PL spectrum of the PDI-O microribbon. Dashed curves are fitted dispersions of polariton modes obtained from the coupled oscillator model. **d**, PL spectrum of the PDI-O microribbon. The dashed lines are fitted Lorentzian line shapes to measure the mode spacing $\Delta\lambda$ for calculation of refractive index. **e**, Refractive index derived from the coupled oscillator model (red line) and calculated from experimental data (black dot). Error bars indicate the 95% confidence intervals propagated from the 95 % confidence intervals for the fitting of the PL spectrum in (**d**).

Comment 3: *I find fig 2a helpful in demonstrating F-P geometry and emission, however, the figure is still a bit dense especially close to zoomed section. I believe authors want to represent conical spread of emission from point dipole and I can figure that reading the manuscript and revision but for first time reader this maybe a bit complicated. Authors should consider making this figure simpler, yet engaging, for the reader.*

Response 3: We appreciate the reviewer's suggestion for improving the representation of the microribbon cavity geometry. According to the reviewer's suggestion, we have modified the Figure 2a in the Manuscript, as shown in Figure R2b in this file, which is more concise and informative.

Response to Reviewer 2

General Evaluation: *The authors present a revised version on their manuscript on polariton condensation in organic microbelt (or in the revised version: microribbon) cavities. In the first round of refereeing three referee reports were returned. The questions raised in all three referee reports were rather similar. So was the general feeling of the referees about whether the manuscript at hand is suitable for Nature Communications: The referees mostly agreed that the topic at hand is suitable for Nature Communications, but the manuscript certainly requires additional clarifications and experimental data. Accordingly, in the following I will mainly focus on the question whether the authors succeeded in clarifying the points raised by the referees. In my opinion, the following points were the most important ones:*

Response: We thank the reviewer for the nice summary of the referees' comment on our work and the evaluation of our responses and revision.

Comment 1: *The experimental geometry*

All referees agreed that the experimental geometry is not described well and may potentially give rise to misunderstandings. Most importantly, a clear description of the propagation direction of polaritons and a clear discussion of which axes are used to record the dispersion (and why) was missing. This point has definitely improved substantially. The explanation that the emission in the x - z -plane is nondirectional due to the small extension of the system along these directions is highly relevant and in my opinion absolutely necessary to understand the manuscript at hand. In my opinion the author response to this point is reasonable and convincing.

Response 1: Thank the reviewer for the recognition of our revision. We also appreciate the suggestions that helps us to improve the demonstration of experimental geometry.

To further clarify the experimental setup and the waveguide cavity geometry, we have

modified Figure 2 in the revised Manuscript, as shown in Figure R3 in this file. Also according to the comments of Reviewer 1, we have added the schematic of angle-resolved PL measurement setup as Figure 2a and modified Figure 2b to be more concise and informative.

Figure R3 **a**, Schematic of the experimental setup for angle-resolved microphotoluminescence (AR μ -PL) measurement. **b**, Schematic of a PDI-O microribbon along with coordinate axes. **c**, AR μ -PL spectrum of the PDI-O microribbon. Dashed curves are fitted dispersions of polariton modes obtained from the coupled oscillator model. **d**, PL spectrum of the PDI-O microribbon. The dashed lines are fitted Lorentzian line shapes to measure the mode spacing $\Delta\lambda$ for calculation of refractive index. **e**, Refractive index derived from the coupled oscillator model (red line) and calculated from experimental data (black dot). Error bars indicate the 95% confidence intervals propagated from the 95 % confidence intervals for the fitting of the PL spectrum in **(d)**.

Comment 2: *validity of polariton condensation*

All referees implied that the authors might be able to find more convincing indications for polariton condensation actually taking place in the sample. The referees suggested

that investigations of the spatial coherence of the emission, a more thorough and quantitative treatment of the spatial redistribution of polaritons due to interactions and a more conclusive input-output curve might be helpful here. In my opinion, the input-output curve and the discussion on the interaction-induced spatial redistribution are convincing now. The additional discussion on spatial coherence is nice, but the contrast observed is not too great. This is somewhat expected due to the typical inhomogeneity of the system. However, as the discussion about the exact experimental geometry was a major point of confusion in the initial manuscript, I would suggest to add axis descriptions and labels to the axes of figures 3)f) and 3)g), so the reader may immediately find the y-direction in the figure. Besides that, I consider the response of the authors with respect to this point satisfactory.

Response 2: We thanks the reviewer for the evaluation and suggestion. We agree that axis labels should be represented in the images of the interference pattern to clarify the microribbon geometry. In Figure 3f and 3g, the direction of the microribbon length is horizontal, and light was emitted from the two lateral sides of the microribbon. Therefore, the vertical and horizontal axis of Figure 3f and 3g correspond to the x- and y-direction, respectively. In order to better represent the mciroribbon geometry, we have added the axis labels to the images of the interference patterns, as shown in the following Figure R4.

Figure R4 The image of the interference pattern in a Young's double-slit interferometry experiment below (a) and above (b) threshold. Scale bars are 8 μm .

According to the reviewer's suggestion, we have replaced the original Figure 3f and

3g with Figure R4a and R4b, respectively, in the revised Manuscript. Accordingly, we have also added a sentence “*The x- and y-direction in (f) and (g) correspond to the direction of the microribbon width and length, respectively.*” in the caption of Figure 3 in the revised version.

Comment 3: *Cavity modes*

There have been several comments from the referees requesting that the authors clarify the mode structure of their cavity modes, both with respect to the quality factor and the polarization properties of the individual modes. In my opinion the authors' response to these points is reasonable.

Response 3: We appreciate for the reviewer’s positive evaluation of our response.

Comment 4: *In my opinion, these were the main points raised. A part of the referees had more specific questions, e.g. on the XRD spectra, where the referee who raised this point will be more competent to judge whether the resply is satisfactory than I am. All referees also seemed to agree that the topic investigated here is relevant and interesting enough for the wider audience of Nature Communications. Accordingly, assuming that the results presented here are free from unintentionally or deliberately introduced errors, I recommend the revised manuscript for publication in Nature Communications.*

Response 4: We thank the reviewer for recognizing the importance of this work and recommending publication of the revised Manuscript. We have further revised the Manuscript according to the comments and suggestions of other reviewers. We wish that these alterations and responses make sense to the reviewers and to the readers.

Response to Reviewer 3

General Evaluation: *The authors made a number of changes to the text and provided reasonable justifications to most of my comments. However, I still have a few issues, particularly with the modeling, which prevent me from recommending publication without further revision.*

Response: We thank the reviewer for pointing out the issues to help us to make a further improvement of our work. In the following, we provide concrete responses to the comments and suggestions point-by-point.

Comment 1: *Generally, I feel that modeling the ribbon as an FP cavity along x is confusing, because it is much more natural to think of the nanoribbons as waveguides along the y -direction, supporting multiple transverse modes (i.e., with characteristic field distributions in the xz -plane). There are many ways to model this - numerical methods are generally necessary to be exact, however simplified methods with semi-analytical expressions also exist (see for instance Marcatili's classical paper on rectangular dielectric waveguides, *The Bell System Technical Journal* Volume: 48, Issue: 7, Sept. 1969) which would be adequate to model the nanoribbons.*

Response 1: Thank the reviewer for the comment about the modeling of the nanoribbon cavity. We understand the confusion pointed out by the reviewer that a nanoribbon waveguide is supposed to support transverse guide modes propagating along the y -direction—a very common situation in waveguide-based devices. According to the suggestion of the reviewer, we have modelled the modes in a nanoribbon dielectric waveguide with a width of 5 μm by solving the eigenvalue equation for the electric field. The simulated electric field distributions on the cross section of the waveguide are shown in the Figure R5a below, from which we can find that the nanoribbon is able to support multiple transverse modes due to its large width. For the modeling of PDI-O single-crystal nanoribbons, we have to take the exciton sources into account, which are modeled as dipoles orienting along y -direction. We

have stimulated the emission profile of a dipole (Figure R5b), which shows that the dipole exhibits negligible emission in the y-direction, while strong emission in the direction perpendicular to the y-direction. Therefore, the emitted light from PDI-O in the nanoribbon is preferentially coupled to the waveguide modes propagating along x-direction, while the transverse modes along y-direction were not effectively excited.

Figure R5 a, Simulated electric field intensity distributions on the cross section in the xy-plane of a nanoribbon with a width of 5 μm . **b**, Stimulated emission profile of a dipole in three-dimensional space (top) and xz or yz-plane (bottom).

To clearly demonstrate the modelling of the nanoribbon waveguide cavity, we have added Figure R5b in the revised Supplementary Information (Figure S10) and the related discussion “*The emission pattern of a dipole is torus-shaped and the emission intensity is strongest in the direction perpendicular to the dipole axis, which indicates that the coupling between the dipole and cavity modes is orientation dependent.*”.

Comment 2: *I also have a bit of an issue with calling the nanoribbons FP cavities in the x-direction, because they extend very far in the y-direction comparatively. In particular, the FP mode simulation discussed in Supplementary Fig. 10 might not be a good representation of the physical structure, because it appears that only a small*

section of a nanoribbon was simulated, and there is no information about the boundary conditions used. This could have artificially selected the TE modes that were presented. The simulation here should be re-run with ‘open boundary’-type boundary conditions, such as Perfectly Matched Layers, or at least Bloch-type boundary conditions.

Response 2: Thanks for the reviewer’s suggestion that would help us to improve our simulation study. According to this suggestion, we have modified our simulation model. The model consists of a $15\ \mu\text{m} \times 2\ \mu\text{m} \times 200\ \text{nm}$ nanoribbon with a dipole source located at its center. The dipole source is parallel to y-direction, and the simulation domain is surrounded by perfectly matched layers (Figure R6a). The simulated results (Figure R6b-d) show that the electric field does not expand in the y-direction due to the directionality of the dipole emission. Instead, the cavity modes in the x-direction are formed in the nanoribbon, which agrees with the experimental results and indicates a large coupling strength between the exciton and the cavity modes in x-direction in the PDI-O single-crystal nanoribbons.

Figure R6 a, Schematic of the simulation model. **b-d**, Simulated electric field intensity distributions on the cross sections of a nanoribbon model in (a).

To avoid confusion and better represent our simulation results, we have replaced the

original Supplementary Figure 10 with Figure R6 in the revised Supplementary Information. Accordingly, we have modified the related discussion as “*This is further supported by the finite-difference time-domain (FDTD) simulation results (Supplementary Fig. 10), which show that for a microribbon with a dipole source perpendicular to its long axis, the cavity modes in the width direction are preferentially formed due to the directionality of the dipole emission.*” and “*Supplementary Fig. 10b shows that when the dipole source is along the y-direction, the microribbon supports the cavity modes in the x-direction due to the directionality of the dipole emission, which suggests a large coupling strength between the dipole and cavity modes in the microribbon. Therefore, in our single-crystal PDI-O microribbons, all the Frenkel excitons can efficiently undergo strong coupling with cavity photons, which facilitates the realization of polariton condensation.*” in the revised Supplementary Information.

Comment 3: *There are also other issues in the responses to some of my original comments:*

Comment 3-1:

*I disagree that k_y can take infinite values, because the ribbon, though long, is not infinite. There should be resonances according to (roughly) $k_y * L = 2 * \pi * n$, with k_y for a particular m , and L the length (extent in the y-direction) of the ribbon. In other words, the ribbon forms a multimode Fabry-Perot resonator along the y-direction. If there are no experimentally visible resonances, perhaps the modal reflectivity at the two ends of the nanoribbon is extremely low, or maybe the spectral resolution is not sufficiently high to resolve such resonances. The authors should verify that that is the case, or offer a better explanation for which such resonances are not observed.*

Response 3-1: Thanks for the comment. Theoretically, for the light modes propagating along y-direction, the wave vector component in y-direction k_y is determined simultaneously by the waveguide characteristic equation

($2k_z - 2\phi_t - 2\phi_b = 2m\pi$, k_z is dependent on k_y and m is integer) and the cavity resonance equation ($k_y = N\pi n/L$, N is integer), leading to discrete islands in the angle-resolved PL spectra. This modulation of PL spectra was not observed in the PDI-O single-crystal nanoribbons, which is ascribed to the orientation of the PDI-O molecules. Because the transition dipole moments of PDI-O molecules are all parallel to the y-direction in the nanoribbons, the emitted light tends to be coupled to the waveguide cavity modes in the x-direction with k_x determined by the width and thickness of the nanoribbon. As shown in Figure R6, for these cavity modes the electric field does not expand to the boundaries in y-direction, and therefore the wave vector component in y-direction k_y is not necessary to satisfy $k_y = N\pi n/L$. This leads to an angle dependent and continuous-valued k_y for each k_x that satisfies simultaneously $2k_z - 2\phi_t - 2\phi_b = 2m\pi$ and $k_x = N\pi n/L$ (k_z is dependent on k_y and m , N are integers).

To avoid confusion, we have added a sentence “*As all the transition dipole moments of PDI-O molecules are parallel to y-direction, the measured PL spectra of the microribbons are only modulated by the waveguide cavity along x-direction (Supplementary Figure 10).*” in the discussion of the y-component of the cavity mode wave vector in the revised Supplementary Information.

Comment 3-2:

Equation (10) in the response is not exact, though it may work as an approximation. In reality, eqs. (11), (12) and (8) have to be solved simultaneously in order to yield a value of β . These expressions, however, are only strictly valid for a slab that extends infinitely in the z direction, which is not the case here. The authors should clarify this in the text. Also, an important simplification that was done in eqs. (11) and (12) was that $n_{air} = n_{sub} = 1$. In reality, the slab has a higher n_{sub} , forming an asymmetric waveguide. This gives a more limited spectrum than the symmetric case. For a self-consistent model, have the authors verified that all the relevant slab modes still exist in the asymmetric structure?

Response 3-2: Thanks for the comment that would help us to increase the rigorousness and validity of our calculation. According to this comment, we have made the following modifications of the analysis and calculation of the waveguide cavity modes:

1) Discussion of the equations:

The propagation constant of the waveguide cavity modes β is simultaneously determined by the finite thickness d (typically ~ 150 nm from AFM measurement result in Supplementary Fig. 5) and width W (several micrometers) of the PDI-O nanoribbons. On one hand, the thickness d determines β through the following equations describing the guided modes propagating along x-direction in the nanoribbon:

$$2k_z d - 2\varphi_t - 2\varphi_b = 2m\pi \quad (1)$$

$$\tan\varphi_t = \frac{\sqrt{\beta^2 - n_{\text{air}}^2 k_0^2}}{k_z} \quad (2)$$

$$\tan\varphi_b = \frac{\sqrt{\beta^2 - n_{\text{sub}}^2 k_0^2}}{k_z} \quad (3)$$

$$k_z = \sqrt{n_{\text{bg}}^2 k_0^2 - \beta^2} \quad (4)$$

These equations were used because the thickness of the nanoribbons ($d \sim 150$ nm) is large enough for the nanoribbons to support guided modes in the spectrum range of the emission of PDI-O.

On the other hand, due to the finite width of the nanoribbons, β also satisfies the equation for an F-P resonator:

$$\beta = \frac{N\pi}{W} \quad (5)$$

Therefore, only the modes with β satisfying these 5 equations simultaneously exist in a PDI-O nanoribbon, and the allowed waveguide cavity modes are dependent on the width and thickness of the nanoribbons.

2) Assumption in the calculation:

The nanoribbon was treated as a symmetric slab waveguide (*i.e.* $n_{\text{air}} = n_{\text{sub}} = 1$) when solving equation (2) and (3) in the original Supplementary Information. According to the reviewer's comment, we have solved the equations using the accurate refractive index of the glass substrate as $n_{\text{sub}} = 1.46$. Figure R7 shows the calculated waveguide and polariton modes, which correspond to Supplementary Figure 13 in the original Supplementary Information and Figure 2b in the original Manuscript, respectively. Although the fitting parameters are slightly different from the fit with assumption of $n_{\text{sub}} = n_{\text{sub}} = 1$, the calculated dispersion curves in Figure R7 that agree well with the measured angle-resolved PL spectra are more accurate to describe the polariton modes in the nanoribbons.

Figure R7 a, Angle-resolved PL spectra at long wavelength of a nanoribbon. Dashed curves are fitted dispersions of uncoupled waveguide cavity modes. **b**, Angle-resolved PL spectra at long wavelength of a nanoribbon. Dashed curves are fitted dispersions of polariton modes obtained from the coupled oscillator model.

According to the reviewer's suggestion, we have added the following sentences in the discussion of the waveguide characteristic equations in the revised Supplementary

Information:

“The thickness of the microribbons is in the range of one to a few hundred nanometers, which is large enough for the microribbon to function as a slab waveguide and support guided modes.”

We have also added the thickness of the nanoribbons in Supplementary Figure 12 and 13 to demonstrate the dependence of polariton branches on both width and thickness of the nanoribbons.

To provide more convincing calculation results, we have modified Figure 2b, Supplementary Figure 11 and 12 by replacing the dashed curves with dispersion curves calculated using the accurate refractive index of the glass substrate as $n_{\text{sub}} = 1.46$. Accordingly, the Rabi splitting was modified as 530 meV in the revised Manuscript. We have also deleted the sentence *“For simplification of calculation, we take $n_{\text{air}} = n_{\text{sub}} = 1$.”* in the revised Supplementary Information.

Comment 3-3:

In Fig. R17 b, the authors plotted error bars indicating 95 % confidence intervals from 5 representative measurements. The error bars to be plotted should be for the refractive index uncertainty, propagated from the 95 % confidence intervals from the fit in Fig. R17a.

Response 3-3: Thanks for pointing out this issue. According to the reviewer’s suggestion, we have re-calculated 95% confidence intervals for the refractive index n by propagating the standard error of peak position σ_λ in Figure 2c into the standard error of refractive index σ_n :

$$n = \frac{\lambda_1 \lambda_2}{2 W(\lambda_2 - \lambda_1)} \quad (6)$$

$$\bar{n} = \frac{\overline{\lambda_1 \lambda_2}}{2 W(\overline{\lambda_2} - \overline{\lambda_1})} \quad (7)$$

$$\sigma_n = \left| \frac{\partial n}{\partial \lambda_1} \right| \sigma_{\lambda_1} + \left| \frac{\partial n}{\partial \lambda_2} \right| \sigma_{\lambda_2} \quad (8)$$

The 95% confidence interval for n is thus

$$[\bar{n} - 1.96\sigma_n, \quad \bar{n} + 1.96\sigma_n] \quad (9)$$

The obtained results are plotted in Figure R8, which shows that the calculated refractive index from the experimental data agrees with that from the coupled oscillator model fitting parameters.

Figure R8 Refractive index derived from the coupled oscillator model (red line) and calculated from experimental data (black dot). Error bars indicate the 95% confidence intervals propagated from the 95 % confidence intervals for the fitting of PL spectrum.

According to the reviewer's suggestion, we have replaced Figure 2d with Figure R6 and modified corresponding caption in the revised Manuscript.

Comment 3-4: *Response to comment 6:*

As in the response to comment 5, the authors should plot the 95 % confidence intervals for the FWHM obtained from the fits to the data.

Response 3-4: We thanks the reviewer for this comment. The measured PL spectra were fitted in OriginLab OriginPro with Lorentzian line shapes for the determination of the PL intensity and FWHM. Figure R9 is a typical fitting result for the PL

spectrum measured at $1.17 P_{th}$, which shows the fitted curves with 95% confidence band as well as standard errors of the PL intensity and FWHM. We have plotted the PL intensity and FWHM as a function of pump fluence in Figure R10, with error bars indicating the 95% confidence intervals obtained from the fits of the experimental data.

Figure R9 Left: PL spectrum of PDI-O nanoribbon measured at $1.17 P_{th}$ (black) and the fitted Lorentzian line shape with 95% confidence band (red). Right: screenshot of the table of fitting parameters for the two peaks in the PL spectrum, which gives the standard errors of the PL intensity (H) and FWHM (w).

Figure R10 Emission intensity and FWHM as a function of pump fluence. The PL intensity was plotted in a log-log scale. Error bars indicate the 95% confidence intervals obtained from the fits to the data.

According to the reviewer's suggestion, we have replaced Figure 3c with Figure R10 and modified corresponding caption in the revised Manuscript. We have also corrected the error bars of Supplementary Figure 13 in the revised Supplementary Information.

Comment 4: Lastly, regarding the vibron-assisted scattering being the reason for the polariton condensation at the 580 nm wavelength, my understanding is that the polariton should be energetically separated from the exciton reservoir by the vibron energy, which is probably of the order of hundreds of meV. I believe the claim in the main text that “The polariton emission is located near 580 nm, which corresponds to $S_{10} \rightarrow S_{01}$ transition of PDI-O”, is not exactly the same. Have the authors characterized the vibronic energies of the PDI-O molecules? I think this information should be provided in the text and better explained, in order to justify the vibron-assisted scattering claim.

Figure R11 Normalized absorption and PL spectra of PDI-O dilute solution.

Figure R12 Relevant Energy levels of PDI-O molecules in the nanoribbon cavity.

Response 4: We appreciate for the comment that would help us to better demonstrate the role of vibrons in the polariton condensation process. From the absorption and PL spectra of PDI-O molecules in Figure R11 (Supplementary Figure 3 in Supplementary Materials) we can find that the absorption peaks are located at 2.36 eV (525 nm) and 2.53 eV (489 nm) corresponding to $S_{00} \rightarrow S_{10}$ transition and $S_{00} \rightarrow S_{11}$ transition respectively. And the emission spectrum is a mirror image of the absorption spectrum

showing the main emission peak at 2.32 eV (535 nm) a vibronic replica at 2.14 eV (580 nm) which correspond to $S_{10} \rightarrow S_{00}$ transition and $S_{10} \rightarrow S_{01}$ transition respectively. The vibronic energy was thus estimated to be $2.32 \text{ eV} - 2.14 \text{ eV} = 180 \text{ meV}$. As shown in Figure R12, the energetic separation between polariton ground state ($k_y = 0$) and PDI-O molecular ground state is 2.14 eV (580 nm, from the polariton emission spectrum) the polariton emission, and therefore the polariton ground state is energetically lower than the exciton reservoir by 180 meV, indicating that the polariton ground state is directly populated with emission of a vibron.

To clearly demonstrate the vibration-assisted scattering process, we have modified the related discussion as “*The polariton emission is located near 580 nm, which corresponds to the energetic separation between exciton reservoir (S_{10}) and the first vibronic sublevel of the molecular ground state (S_{01}), indicating molecular vibration-assisted population of polaritons from the exciton reservoir.*” in the revised Manuscript.

REVIEWER COMMENTS

Reviewer #1 (Remarks to the Author):

The authors have done a good job on improving the figures better and FDTD simulations does make it easier for reader to visualise dipole and field distribution in the PDI-O. They have satisfactorily addressed my concerns and I'm happy to recommend the publication in Nat. Comm noting one minor edit to revised text added line 4 pg 8 where authors report high Rabi splitting observed in PDI crystals. It would be compare these values against reported Rabi splitting in crystals such as anthracenes and other PDIs - if I remember correctly, latter have mostly been on thin films and not crystals but worth a check.

Reviewer #2 (Remarks to the Author):

The authors have resubmitted a revised version of their manuscript. In the last round of refereeing, two referees had minor comments and recommended publication and the third referee pointed towards some issues with modeling the data, which the authors should solve before the manuscript can be published.

In my opinion the authors' response to the first two referees is reasonable and sound. Referee 3 requested an extension of the underlying model, such that the asymmetry of the waveguide due to the different refractive indices is considered and a more detailed reasoning about why the x-axis of the microbelt plays a distinguished role as compared to the y-axis.

The first of the questions certainly has been answered satisfactorily. The refined model shows reasonable agreement with experimental data. From my point of view, also the second point has been answered reasonably well by the authors in the response to the referee: the preferred orientation of dipoles is responsible for this. However, it is my impression that this point has not been clarified fully in the manuscript. In fact, I think this is a continuation of the problem that was mentioned already in the first round of referee reports: the presentation of the experimental geometry might not be immediately clear to a reader who is not familiar with the system at hand. Figures 2)a) and b) now clearly define in which direction the x-, y- and z-axes point, which intuitively explains in which direction the pump and the microbelt dimensions point. However, as the remark from referee 3 shows, the other relevant directions in the system are not as clear, most notably the dipole orientation. It would make the manuscript much easier to read if one could also identify these directions immediately. My suggestion would be to include a depiction of the x-, y- and z- axes also in figure 1 - most importantly in 1)d) and 1)e), but possibly also in 1)b). By doing so and mentioning that the [010] axis corresponds to the y-axis, which in turn corresponds to the mean orientation of the dipole moments, it should become possible to understand the experimental geometry in a much more intuitive manner.

However, I think this is a point of presentation and it can be added quite easily. Apart from this point, I think the manuscript should be published once the point mentioned above has been added.

Reviewer #3 (Remarks to the Author):

1 - Please double-check the axes in Fig. R1b (Supplementary Fig. 10b). On the top right figure, it looks like the y-axis should be either z or x, and likewise with the bottom left figure. Please label the axes with numbers for all the dimensions. – the aspect ratio of the figure may change, but the reader will be able to verify which axis is the long one, and which ones are for the cross-section. This must absolutely be done.

Because it's not possible to understand what fields are being shown here, the response to my comment 3 is not acceptable. In particular, even if the emitter dipoles are oriented in the y-direction, there will be coupling to waveguide modes that travel in the y-direction. See for instance Stepanov et al., Appl. Phys. Lett. 106, 041112 (2015), where a longitudinal (y-oriented) dipole excites the mode in the bottom of Fig. 1(b), which travels in the y-direction. While the refractive indices are different there, dipole coupling will take place likewise here.

This discussion affects the interpretation of the results, however. I just think the explanation ("As all the transition dipole moments of PDI-O molecules are parallel to y-direction, the measured PL spectra of the microribbons are only modulated by the waveguide cavity along x-direction (Supplementary Figure 10).") is not appropriate, so I would just get rid of it.

2 - I think the discussion regarding Fig. 5a, and which waveguide modes the y-dipole excites is incorrect. My understanding is that the calculated waveguide modes travel into the page, with a transverse field distribution that corresponds to a standing-wave pattern on the plane of the figure. The 15 μm length of the nanoribbon plays no role here, only the width and the thickness. If the plane here is xy, then the situation does not correspond to the axes shown in Fig. R1b. I don't think an extended discussion is necessary here, as it does not affect the results. I would be happy to just see the axes are properly labeled in Suppl. Fig. 10b.

3- Thank you for the clarification regarding the exciton reservoir separation. I suggest adding Figure R12 to the SI or to the main text somewhere, to clarify the explanation for the reader.

Response to Reviewer 1

General Evaluation: *The authors have done a good job on improving the figures better and FDTD simulations does make it easier for reader to visualise dipole and field distribution in the PDI-O.*

They have satisfactorily addressed my concerns and I'm happy to recommend the publication in Nat. Comm. noting one minor edit to revised text added line 4 pg 8 where authors report high Rabi splitting observed in PDI crystals. It would be compare these values against reported Rabi splitting in crystals such as anthracenes and other PDIs - if I remember correctly, latter have mostly been on thin films and not crystals but worth a check.

Response: We thank the reviewer for the positive evaluation of our revision and recommendation for publication of the revised Manuscript. We also appreciate the reviewer's suggestion for providing more information about Rabi splitting in organic crystals. We have reviewed the previous works on strong coupling in microcavities containing similar materials to PDI-O and found that the reported Rabi splitting (Ω) ranges from tens to a few hundred meV. For instance, $\Omega = 100$ meV in a half-VCSEL microcavity embedding BP1T-CN crystal (*Appl. Phys. Lett.* 109, 061101 (2016)), $\Omega = 200$ meV in a DBR-based microcavity containing anthracene crystal (*Phys. Rev. Lett.* 101, 116401 (2008)), and $\Omega = 150$ meV in all metal microcavities that consist of PDI-doped polymer film (*J. Mater. Chem. C* 7, 2954 (2019)). Compared with these microcavities, our PDI-O microribbons exhibit a larger Rabi splitting of $\Omega = 530$ meV, which is attributed to the high molecular density in the crystal, the large transition dipole moment of PDI-O molecule and the maximum overlap between the transition dipole moment and the electric field in the microcavity (*Chem. Soc. Rev.* 48, 937 (2019)).

According to the reviewer's suggestion, we have modified the related discussion as "*Rabi splitting of $\Omega = 530$ meV extracted from the fitted data indicates a very large coupling strength, which is resulted from the high molecular density in the crystal, the*

large transition dipole moment of PDI-O molecule, and the maximum overlap between the transition dipole moment and the electric field in the microcavity^{31,32,35,38} and included the above-mentioned references in the revised Manuscript.

Response to Reviewer 2

General Evaluation: *The authors have resubmitted a revised version of their manuscript. In the last round of refereeing, two referees had minor comments and recommended publication and the third referee pointed towards some issues with modeling the data, which the authors should solve before the manuscript can be published. In my opinion the authors' response to the first two referees is reasonable and sound. Referee 3 requested an extension of the underlying model, such that the asymmetry of the waveguide due to the different refractive indices is considered and a more detailed reasoning about why the x-axis of the microbelt plays a distinguished role as compared to the y-axis. The first of the questions certainly has been answered satisfactorily. The refined model shows reasonable agreement with experimental data. From my point of view, also the second point has been answered reasonably well by the authors in the response to the referee: the preferred orientation of dipoles is responsible for this.*

Response: We thank the reviewer for the nice summary and the positive evaluation of our responses and revision.

Comment 1:

However, it is my impression that this point has not been clarified fully in the manuscript. In fact, I think this is a continuation of the problem that was mentioned already in the first round of referee reports: the presentation of the experimental geometry might not be immediately clear to a reader who is not familiar with the system at hand. Figures 2)a) and b) now clearly define in which direction the x-, y- and z-axes point, which intuitively explains in which direction the pump and the microbelt dimensions point. However, as the remark from referee 3 shows, the other relevant directions in the system are not as clear, most notably the dipole orientation. It would make the manuscript much easier to read if one could also identify these directions immediately. My suggestion would be to include a depiction of the x-, y- and z- axes

also in figure 1 - most importantly in 1)d) and 1)e), but possibly also in 1)b). By doing so and mentioning that the $[010]$ axis corresponds to the y -axis, which in turn corresponds to the mean orientation of the dipole moments, it should become possible to understand the experimental geometry in a much more intuitive manner. However, I think this is a point of presentation and it can be added quite easily.

Response 1: We are grateful for the reviewer's valuable suggestion that helps us to further improve the presentation of the microribbon cavity geometry. The addition of the depiction of axes in Figure 1d and 1e would not only highlight the dipole orientation in the microribbon but also help the readers to identify the coordinate system in this work soon, which makes it easier to understand the experimental setup and the cavity geometry. According to this suggestion, we have added axes in Figure 1d and 1e, as shown in Figure R1, which defines the coordinate system and provides a guidance of the dipole direction (y -axis). In addition, we have added red double-headed arrows denoting the dipoles in Figure 1e (Figure R1), 2a and 2b (Figure R2) to further clarify the experimental geometry.

Figure R1 PDI-O single-crystal microribbon cavities for polariton Bose-Einstein condensation. **a**, Chemical structure of the PDI-O molecule. **b**, SEM image of a PDI-O microribbon. Scale bar: 25 μm . **c**, TEM image and corresponding SAED patterns of a PDI-O microribbon. Scale bars are 5 μm and 2 $1/\text{nm}$, respectively. **d**,

Spatial relationship between the PDI-O molecular transition dipole moment (red double-headed arrow) and the [010] growth direction (black arrow) of the PDI-O microribbon. The dipole direction is along the ribbon length, which is defined as y-axis. **e**, Schematic illustration of polariton condensation in an organic microcrystal cavity. LPB: lower polariton branch.

Figure R2 a, Schematic of the experimental setup for AR μ -PL measurement. **b**, Schematic of a PDI-O microribbon along with coordinate axes. The waveguide F-P cavity direction is parallel to the x-axis direction. k_y is the projection of wave vector on to y-axis, which depends upon the emission angle in y-z plane (θ).

According to the reviewer's suggestion, we have replaced Figure 1 with Figure R1 and added a sentence "The dipole direction is along the ribbon length, which is defined as y-axis." in the caption of Figure 1 in the revised Manuscript. We have also replaced Figure 2a and 2b with Figure R2a and R2b, respectively.

Comment 2: *Apart from this point, I think the manuscript should be published once the point mentioned above has been added.*

Response 2: We thank the reviewer for recommending publication of the revised Manuscript.

Response to Reviewer 3

Comment 1: *Please double-check the axes in Fig. R1b (Supplementary Fig. 10b). On the top right figure, it looks like the y-axis should be either z or x, and likewise with the bottom left figure. Please label the axes with numbers for all the dimensions. – the aspect ratio of the figure may change, but the reader will be able to verify which axis is the long one, and which ones are for the cross-section. This must absolutely be done.*

Because it's not possible to understand what fields are being shown here, the response to my comment 3 is not acceptable. In particular, even if the emitter dipoles are oriented in the y-direction, there will be coupling to waveguide modes that travel in the y-direction. See for instance Stepanov et al., Appl. Phys. Lett. 106, 041112 (2015), where a longitudinal (y-oriented) dipole excites the mode in the bottom of Fig. 1(b), which travels in the y-direction. While the refractive indices are different there, dipole coupling will take place likewise here.

This discussion affects the interpretation of the results, however. I just think the explanation (“As all the transition dipole moments of PDI-O molecules are parallel to y-direction, the measured PL spectra of the microribbons are only modulated by the waveguide cavity along x-direction (Supplementary Figure 10).”) is not appropriate, so I would just get rid of it.

Response 1: We thank the reviewer for the suggestion and comment about the simulation. According to the suggestion of the reviewer, we have double-checked the axes in Supplementary Fig. 10b and added axis labels to clarify the planes where we put the monitors to obtain the electric field distribution in FDTD solutions, as shown in Figure R3. The top right panel in Figure R3 shows the electric field distribution in the xy-plane, from which we consider that the dipole emission does not couple into the waveguide modes in the y-direction. Instead, the electric field distributions in the xz and yz-planes indicate that the waveguide mode in the x-direction is formed in the microribbon (the bottom panels in Figure R3). The PL image of a microribbon excited

by a laser spot shows bright emission from both lateral facets (Figure R4), indicating that the simulations coincide well with the experimental results, which convinced us that the measured spectra from our PDI-O microribbons correspond to the cavity modes in the x-direction. This does not necessarily yield the conclusion that the emission of the dipole oriented in the y-direction only couples to the waveguide modes in the x-direction, and therefore we agree with the reviewer that the claim “*As all the transition dipole moments of PDI-O molecules are parallel to y-direction, the measured PL spectra of the microribbons are only modulated by the waveguide cavity along x-direction (Supplementary Figure 10).*” is inappropriate. We also appreciate the reviewer for recommending the work of Stepanov *et al.* (*Appl. Phys. Lett.* 106, 041112 (2015)), which is a good reference for the Manuscript because it supports the argument that coupling to waveguide modes are strongly dependent on the dipole orientation, as evidenced by their simulation results in their Fig. 1c (Figure R5 in this file).

Figure R3 Simulated electric field intensity distribution in a microribbon with the dipole source μ (red double-headed arrow) located at its center ($x = 1 \mu\text{m}$, $y = 7.5 \mu\text{m}$, $z = 100 \text{ nm}$). The dipole is parallel to the y-direction.

Figure R4 PL image of a microribbon excited by a laser spot. Scale bar: 10 μm .

Figure R5 The model and simulation results in the work by Stepanov *et al.* (*Appl. Phys. Lett.* 106, 041112 (2015)). (a) Sketch of the ridge waveguide system and geometry of the optics experiments. The (X, Y, Z) axis system corresponds to quantum dot crystalline eigenaxes and (\perp , \parallel) to the photonic eigenaxes. (b) Spatial intensity profile of the fundamental guided mode M calculated for $h = 130$ nm, $\omega = 300$ nm, and $\lambda = 950$ nm. E_{\perp} (E_{\parallel}) is the longitudinal (transverse) electric field component associated with M. (c) Calculated normalized spontaneous emission rates plotted against the reduced wire width w/k for linear optical dipoles, oriented along \perp and \parallel . The wire is infinitely long and the dipole is located on the wire axis.

According to the comment and suggestion of the reviewer, we have replaced Supplementary Fig. 10b with Figure R3 in the revised Supplementary Information. We have also deleted the sentence “As all the transition dipole moments of PDI-O

molecules are parallel to y-direction, the measured PL spectra of the microribbons are only modulated by the waveguide cavity along x-direction (Supplementary Figure 10)." and modified the related discussion as *"For the waveguide cavity modes in the x-direction, because of the long length of the microribbon, the y-component of the wave vector k_y is free and can be expressed by..."* in the revised Supplementary Information. The above-mentioned reference (*Appl. Phys. Lett.* 106, 041112 (2015)) has also been added in the revised Manuscript.

Comment 2: *I think the discussion regarding Fig. 5a, and which waveguide modes the y-dipole excites is incorrect. My understanding is that the calculated waveguide modes travel into the page, with a transverse field distribution that corresponds to a standing-wave pattern on the plane of the figure. The 15 um length of the nanoribbon plays no role here, only the width and the thickness. If the plane here is xy, then the situation does not correspond to the axes shown in Fig. R1b.*

I don't think an extended discussion is necessary here, as it does not affect the results. I would be happy to just see the axes are properly labeled in Suppl. Fig. 10b.

Response 2: Thanks for the reviewer's comment. Although it does not affect the results, we think it is necessary to further clarify the Figure R5 in the last-round revision. The distributions of electric field intensity on the cross section in the xz-plane of a microribbon are shown in Figure R6 in this file. The electric field patterns correspond to the transverse waveguide modes traveling along the y-direction, which were obtained by solving the eigenvalue equation for the electric field without considering the dipole source.

Figure R6 a, A microribbon with width of 5 μm and thickness of 200 nm. **b**, Simulated electric field intensity distributions on the cross section in xz -plane of the microribbon.

According to the reviewer's suggestion about the axis labels in Supplementary Figure 10b, we have added the axis labels in the panels depicting the electric field profiles, as shown in Figure R3 and discussed in Response 1. These axis labels help to clarify the planes where the electric field intensity distributions on the cross section are displayed.

Comment 3 *Thank you for the clarification regarding the exciton reservoir separation. I suggest adding Figure R12 to the SI or to the main text somewhere, to clarify the explanation for the reader.*

Response 3: We appreciate the reviewer's recognition of our revision and the suggestion for clarifying the energetic separation between polariton ground state and PDI-O molecular ground state to better demonstrate the vibron-assisted scattering process. Following this suggestion, we have added the relevant energy levels of PDI-O molecules (Figure R7) in the revised Supplementary Information as Supplementary Fig. 12.

Figure R7 Relevant energy levels of PDI-O molecules in the microribbon cavity.

Accordingly, we have also added following discussion in the revised Supplementary Information:

The polariton emission is located near 580 nm, which indicates that the energetic separation between polariton ground state ($k_y = 0$) and PDI-O molecular ground state (S_{00}) is 2.14 eV. This value equals to the energetic separation between exciton reservoir (S_{10}) and the first vibronic sublevel of the molecular ground state (S_{01}), indicating molecular vibration-assisted population of polaritons from the exciton reservoir, as shown in Supplementary Fig. 12.

From the absorption and PL spectra of PDI-O molecules in Supplementary Fig. 3 we can find that the absorption peaks are located at 2.36 eV (525 nm) and 2.53 eV (489 nm), corresponding to the $S_{00} \rightarrow S_{10}$ and $S_{00} \rightarrow S_{11}$ transitions respectively. And the emission spectrum is a mirror image of the absorption spectrum showing the main emission peak at 2.32 eV (535 nm) and a vibronic replica at 2.14 eV (580 nm) which correspond to the $S_{10} \rightarrow S_{00}$ and $S_{10} \rightarrow S_{01}$ transitions respectively. The vibronic energy was thus estimated to be $2.32 \text{ eV} - 2.14 \text{ eV} = 180 \text{ meV}$. As shown in Supplementary Fig. 12, the energetic separation between polariton ground state ($k_y = 0$) and PDI-O molecular ground state (S_{00}) is 2.14 eV (580 nm, from the polariton emission spectrum), and therefore the polariton ground state is energetically lower than the exciton reservoir by 180 meV, indicating that the polariton ground state is directly populated with emission of a vibron.

REVIEWER COMMENTS

Reviewer #3 (Remarks to the Author):

Thank you for adding axis labels to the figures as I had requested. I unfortunately think that there is still not enough information in the text for one to understand what is taking place. To be clear, though - I don't think that the point the authors are trying to make is incorrect, I just think that the simulation they provide to illustrate the point needs more clarification. The reasons are as follows.

I appreciate that the direction of the dipole determines what modes get excited, and in fact that was something that was shown in APL paper I suggested. However, this was only one of the points that required clarification regarding the simulation. Now that the axes are properly labeled, the following point should be clarified:

1 - Why does the dipole seemingly couple only to one modal order? The waveguide is highly multimode, having index >3 and 2 μm in width, and must support a large family of modes, many of which would couple to the longitudinal dipole - giving a highly complex standing wave pattern. Note that this highly multimode situation contrasts greatly with the case of the APL 2015 paper, in which there was only one higher-order mode that coupled to the longitudinal dipole. The attached simulation shows what one would expect in the present case. In the simulation, the waveguide has an index $n = 3.0$, width 2 μm and length 5 μm . A longitudinal (y-oriented) dipole at the waveguide center excites it. It is clear from the shown complex standing-wave patterns that modes of many different transversal orders have been excited by the dipole. In particular, the formed standing-wave patterns have strong field modulation caused by the interference of the multiple excited modes. How is this situation different than the one in the authors' simulation? This is not clear in the paper.

2 - Why does the standing wave in Fig. S10 have only one anti-node in the y-direction? If this is proper confined mode, it should anti-nodes close to $k_y y = m\pi$. Is this a mode with k_y tending to zero? In this case, the mode would surely be above the light line, meaning that it would not be well confined in the waveguide (i.e., it would be leaky, and would extend into the top and bottom cladding). In Figs. S10, the cross-section field plots on the yz and xz planes should show what happens above and below the PDI-O as well (currently, the plotting region is limited to the PDI-O only).

One possibly relevant point about this is that since the vertical confinement is not great for waves near $k_y=0$, the dipolar coupling is likely not great. This type of waveguide is not the same thing as a microcavity with DBR mirrors, because the field confinement in that case is not based on total internal reflection, so there's strong confinement even for small-k modes. So it's in a way a bit surprising that strong coupling can be observed near $k_y=0$ here.

3 - What refractive index was used? Was it a dispersive index, to better model the material? If so, what method was used to model it in the FDTD code?

Again, this is mostly so things are clear as far as the simulation goes - the main point, that the y-dipole excites modes that give a primarily y-polarized far-field, seems ok to me. However if the authors want to keep the simulations in the paper, all of the comments above must be properly addressed.

Response to Reviewer 3

Comment: *Thank you for adding axis labels to the figures as I had requested. I unfortunately think that there is still not enough information in the text for one to understand what is taking place. To be clear, though - I don't think that the point the authors are trying to make is incorrect, I just think that the simulation they provide to illustrate the point needs more clarification. The reasons are as follows.*

I appreciate that the direction of the dipole determines what modes get excited, and in fact that was something that was shown in APL paper I suggested. However, this was only one of the points that required clarification regarding the simulation. Now that the axes are properly labeled, the following point should be clarified:

1 - Why does the dipole seemingly couple only to one modal order? The waveguide is highly multimode, having index >3 and 2 μm in width, and must support a large family of modes, many of which would couple to the longitudinal dipole - giving a highly complex standing wave pattern. Note that this highly multimode situation contrasts greatly with the case of the APL 2015 paper, in which there was only one higher-order mode that coupled to the longitudinal dipole. The attached simulation shows what one would expect in the present case. In the simulation, the waveguide has an index $n = 3.0$, width 2 μm and length 5 μm . A longitudinal (y-oriented) dipole at the waveguide center excites it. It is clear from the shown complex standing-wave patterns that modes of many different transversal orders have been excited by the dipole. In particular, the formed standing-wave patterns have strong field modulation caused by the interference of the multiple excited modes. How is this situation different than the one in the authors' simulation? This is not clear in the paper.

2 - Why does the standing wave in Fig. S10 have only one anti-node in the y-direction? If this is proper confined mode, it should have anti-nodes close to $k_y y = m\pi$. Is this a mode with k_y tending to zero? In this case, the mode would surely be above the light line, meaning that it would not be well confined in the waveguide (i.e., it would be leaky, and would extend into the top and bottom cladding). In Figs. S10, the cross-section field plots on the yz and xz planes should show what happens above and below the PDI-O as well (currently, the plotting region is limited to the PDI-O only).

One possibly relevant point about this is that since the vertical confinement is not great for waves near $k_y=0$, the dipolar coupling is likely not great. This type of waveguide is not the same thing as a microcavity with DBR mirrors, because the field confinement in that case is not based on total internal reflection, so there's strong confinement even for small- k modes. So it's in a way a bit surprising that strong coupling can be observed near $k_y=0$ here.

3 - What refractive index was used? Was it a dispersive index, to better model the material? If so, what method was used to model it in the FDTD code?

Again, this is mostly so things are clear as far as the simulation goes - the main point, that the y -dipole excites modes that give a primarily y -polarized far-field, seems ok to me. However if the authors want to keep the simulations in the paper, all of the comments above must be properly addressed.

Response: We are very grateful for the reviewer's enthusiasm and patience in helping us to improve the simulation part of our work. We have realized that although we have made significant progresses with the help of the reviewer's suggestions and comments in the past few rounds of revisions, there remains some ambiguity in the lateral confinement of modes in the microribbon.

First, to clearly demonstrate the simulation work for better supporting the experimental results, we have carefully checked the whole manuscript and provided a clarified elucidation of what is going on in the PDI-O microribbons as following:

The prepared PDI-O microribbon exhibits uniform width (W) of several micrometers and thickness (d) in the range of one to a few hundred nanometers along the entire length (L) of several hundred micrometers (Figure R1a). Such a structure is able to function as a waveguide, where light is confined in the z -direction and travels in the x and y -directions. In addition, due to the finite width and length, the guided light may also be reflected back and forth, forming cavity modes in the x or y -direction. To analyze the waveguide cavity modes, we excited the microribbon with a laser spot and measured the PL spectrum. Figure R1b is the PL image of a typical microribbon,

which shows strong emission near the excitation area, suggesting that light does not travel very far along the y-direction. The measured angle-resolved PL spectra (Figure R1c, Figure 2c in the manuscript) exhibit multiple discrete dispersion curves, which agree well with the cavity modes in the x-direction ($k_x W = m\pi$). Therefore, the experimental results indicate that upon laser pumping, the cavity modes in the x-direction are preferentially formed in the PDI-O microribbon.

Figure R1 **a**, Schematic of the prepared PDI-O microribbon. **b**, PL image of a microribbon excited by a laser spot. Scale bar: 10 μm . **c**, Angle-resolved PL spectra of the PDI-O microribbon.

Here, we provide the point-to-point responses to the reviewer's questions:

1 - About the high-order modes along the y direction.

We thank the reviewer for providing the detailed numerical simulation of the modes in a microribbon, which manifests that modes of many different transversal orders can be formed in the waveguide. We agree with the reviewer that there are high-order modes in a microribbon structure with uniform refractive index distribution. Here, the

situation of our experimental device is different from the ideal model, because the refractive index distribution of the microribbon would be changed by the external pump field. We understand that it is surprising to observe the localization of mode in a microribbon structure, which has translational symmetry in y-direction. It should be mentioned that similar results of localized ultra-high quality factor modes have been reported in a cylinder structure, where the localization is attributed to the perturbation of translational symmetry by an external nanofiber coupler or a nanobump of the surface (*Optics Letters* **35**, 2385 (2010), *Opt. Express* **19**, 26470 (2011)).

In this work, the observed mode localization is also resulted from the broken translational symmetry of the microribbon by the pump induced non-uniform refractive index distribution. The perturbation can be attributed to two distinct mechanisms:

(1) Under the pump laser excitation, the energy deposited by the laser may lead to the variations of the material density and thus an increase of the refractive index at excitation area (*Opt. Lett.* **33**, 651-653 (2008)). In addition, a large fraction of PDI-O molecules is pumped into the excited states, and the significant population redistribution would produce an additional increase in the refractive index (*Nat. Commun.* **6**, 8420 (2015)). The photoinduced refractive index increase results in weak potential for light confinement in the y-direction.

(2) In the pump region, the light is generated and amplified. However, outside the pump region, the re-absorption of PDI-O molecules induces optical loss in the waveguide, and the light propagating along y-direction suffers larger loss than that localized in the pump region.

To verify the mode localization in the microribbon due to the above two mechanisms, we have modified the model and re-run the simulation in FDTD solutions. We performed numerical simulation of optical field in the microribbon, with the central area has a higher refractive index ($2.3 > 2.2$) and no optical absorption loss $\text{Im}(n)=0$, as

shown in Figure R2. From the simulated results we can observe that the light field is largely confined in the y-direction, forming cavity modes in the x-direction. We believe these simulation results are reasonable for supporting the experimental results in this work.

Figure R2 Simulated electric field intensity distribution in a microribbon with the dipole source (red double-headed arrow) located at the center ($x = 1 \mu\text{m}$, $y = 7.5 \mu\text{m}$, $z = 100 \text{ nm}$). The dipole is parallel to the y-direction. The refractive index near the dipole source is set to be $n = 2.3$, which is slightly larger than the background value (2.2) considering the thermal and excited-state refractive index changes under laser excitation. The refractive index outside the excitation area is taken as $n = 2.2 + 0.01i$ for the introduction of loss due to re-absorption of PDI-O molecules. The white dashed squares indicate the profile of the microribbon.

2 - node in the y-direction.

The absence of nodes in the y-direction is closely related to the mode localization in the y-direction. As explained above, the weak perturbation of refractive index along

the y-direction would create a weak confinement potential for optical fields, and therefore the localized mode features $k_y \sim 0$. The confinement of the modes with anti-nodes are much weaker due to their large k_y , which results in lower quality factors of these modes than that of the localized mode with $k_y \sim 0$. When the microribbon was pumped by a laser, the modes with highest quality factors are preferentially formed, and as a result the anti-nodes in the y-direction were not observed.

The $k_y \sim 0$ does not necessarily mean that the light could not be well confined in the waveguide. We provide the electric field intensity distributions in the top and bottom cladding in Figure R2, which show that the optical mode does not leak outside the PDI-O microribbon. The light with $k_y \sim 0$ can also be strongly confined because according to $k_x^2 + k_y^2 = k_{xy}^2$, k_x is large when $k_y \sim 0$. As shown in the Fig. R2, the field in x-direction shows multiple anti-nodes and confirms the large k_x .

3 - About the refractive index used in the model.

In the simulation, considering the relatively narrow wavelength range (from 570 nm to 590 nm), we used a fixed refractive index that is independent on wavelength. The background refractive index of the microribbon was set to be $2.2 + 0.01i$, with the imaginary part introducing a loss resulted from re-absorption of PDI-O molecules. The refractive index at central area is slightly higher than the background value (2.3) and the imaginary part was removed considering that light is generated and amplified upon laser pumping. Because the microribbon is excited with a dipole source, there exist short-lived transients resulted from the strong emission of the dipole near the start of the simulation. To filter away these unwanted transients and get the correct mode profile for the cavity, we used a time apodization (start apodization, center: 20 fs, width: 2 fs) in the Monitor (see the description for apodization option of Frequency monitors in the official website of Lumerical: <https://support.lumerical.com>).

According to the above comments by the reviewer, we have added the figures and discussions to clarify the simulation results: added Figure R1b in Supplementary Figure 10, and replaced Supplementary Figure 10b with Figure R2 in the revised Supplementary Information. We have also modified the related discussion as following:

When excited with a laser spot, the PDI-O microribbon exhibits bright emission from both lateral facets (Supplementary Fig. 10a), manifesting that the cavity modes in the ribbon width direction (defined as x-direction) are preferentially excited. The absence of cavity modes in the ribbon length (defined as y-direction) is attributed to (i) the directionality of the dipole emission (Supplementary Fig. 10b), (ii) the large optical loss induced by re-absorption of the PDI-O molecules during the long-distance propagation of light in the y-direction, and (iii) the weak light confinement resulted from the refractive index increase at the excitation area. To verify this conclusion, we have performed numerical simulation of the electric field intensity distributions in the microribbon. The microribbon with a background refractive index $n_{bg} = 2.2 + 0.01i$ was excited with a dipole source located at the center, and the refractive index near the source was set to be slightly higher than n_{bg} ($n = 2.3$). Note that the imaginary part of n_{bg} indicates the absorption-induced optical loss. Supplementary Fig. 10c shows the simulated electric field intensity distributions in the microribbon, from which we can find that light is confined in the y-direction and coupled into cavity modes in the x-direction.

The PDI-O microribbon excited by a laser spot exhibits bright emission from both lateral facets, indicating that the emission from PDI-O molecules (dipole sources) is coupled into the cavity modes in the width direction. The emission pattern of a dipole is torus-shaped and the emission intensity is strongest in the direction perpendicular to the dipole axis, which indicates that the coupling between the dipole and cavity modes is orientation dependent. Under the pump laser excitation, the energy

deposited by the laser may lead to the variations of the material density and thus an increase of the refractive index at excitation area⁷. In addition, a large fraction of PDI-O molecules are pumped into the excited states, and the significant population redistribution would produce an additional increase in the refractive index⁸. The photoinduced refractive index increase further results in weak potential for light confinement in the y-direction. The simulated electric field intensity distributions (Supplementary Fig. 10c) show that the emission of the dipole was confined in the y-direction and coupled into cavity modes in the x-direction, which agrees well with the experimental results (Supplementary Fig. 10a). Therefore, the generated Frenkel excitons in the single-crystal PDI-O microribbons can efficiently undergo strong coupling with cavity photons, which facilitates the realization of polariton condensation.

We have added the following specific description of our numerical model in the Method in the revised Supplementary Information:

The electric field intensity distribution in a single microribbon was calculated by finite-difference time-domain (FDTD) simulation (FDTD solutions, Lumerical solutions, Inc). The model consists of a microribbon ($15\ \mu\text{m} \times 2\ \mu\text{m} \times 200\ \text{nm}$) with a dipole source located at the center. The dipole source is parallel to the ribbon length, and the simulation domain is surrounded by perfectly matched layers. The background refractive index of the microribbon was set to be $2.2 + 0.01i$, with the imaginary part introducing a loss resulted from re-absorption of the PDI-O molecules. The refractive index at the center area is slightly higher than the background value (2.3), and the imaginary part was removed considering that light is generated and amplified upon laser pumping. As the system is excited with a dipole source, a time apodization (start apodization, center: 20fs, width: 2 fs) was used to filter away the unwanted transients near the start of the simulation and thus get the correct mode profile.

We have also added the above-mentioned references (*Opt. Lett.* **33**, 651-653 (2008),

Nat. Commun. **6**, 8420 (2015)) as Refs 7 and 8 in the revised Supplementary Information.

REVIEWERS' COMMENTS

Reviewer #3 (Remarks to the Author):

I am satisfied with the authors' clarification that the spot illumination ultimately defines the waveguide boundaries. This point should be well clarified in the manuscript to avoid confusion. This is akin to gain-guiding in optically pumped lasers which is common practice in organic optoelectronics. I am happy to recommend publication in Nature Comms.